# The translation inhibitors kasugamycin, edeine and GE81112 target distinct steps during 30S initiation complex formation

Haaris A. Safdari [1,5], Martino Morici[1,5], Ana Sanchez-Castro [2,5], Andrea Dallapè[1,3], Helge Paternoga [1], Anna Maria Giuliodori[4], Attilio Fabbretti [4], Pohl Milón [2] ✉ & Daniel N. Wilson [1] ✉

During bacterial translation initiation, the 30S ribosomal subunit, initiation factors, and initiator tRNA define the reading frame of the mRNA. This process is inhibited by kasugamycin, edeine and GE81112, however, their mechanisms of action have not been fully elucidated. Here we present cryo-electron microscopy structures of 30S initiation intermediate complexes formed in the presence of kasugamycin, edeine and GE81112 at resolutions of 2.0-2.9 Å. The structures reveal that all three antibiotics bind within the E-site of the 30S and preclude 30S initiation complex formation. While kasugamycin and edeine affect early steps of 30S pre-initiation complex formation, GE81112 stalls pre-initiation complex formation at a further step by allowing start codon recognition, but impeding IF3 departure. Collectively, our work highlights how chemically distinct compounds binding at a conserved site on the 30S can interfere with translation initiation in a unique manner.

Translation initiation is the first phase of protein synthesis where the start codon of the mRNA is recognized by the initiator fMet-tRNA[fMet] and, as such, is a critical step for selecting the correct open reading frame of the mRNA[1–3]. In bacteria, translation is facilitated by three conserved initiation factors, IF1, IF2 and IF3, which play essential roles in the efficiency and fidelity of the process[4]. The first step during bacterial translation initiation is the formation of a 30S pre-initiation complex (30S-PIC), which comprises the 30S ribosomal subunit, mRNA, initiator tRNA and all three initiation factors[5]. Upon correct decoding of the start codon by the initiator tRNA, the 30S head shifts from an open to a closed conformation[6], and the 30S-PIC transitions to a 30S initiation complex (30S-IC)[5]. Formation of 30S-IC involves conformational rearrangements that increase the IF3 dissociation rate (or at least the C-terminal domain (CTD) of IF3) from its binding site near the P-site, as well as an affinity increase of IF1, IF2 and the initiator tRNA (refs. 5–8). The dissociation of IF3 relieves a steric hindrance that

enables association of the 50S subunit with the 30S-IC, which in turn activates the GTPase activity of IF2. Once the initiator tRNA is accommodated at the peptidyl-transferase center (PTC) of the 50S, IF1 and IF2-GDP dissociate and the 70S-IC becomes elongation competent[1,3,7,9].

Translation is a major target for antibiotics and compounds that inhibit each step of the process, including translation initiation, have been reported[10–12]. Well-known inhibitors of translation initiation include natural product compounds, such as kasugamycin (Ksg), edeine (Ede) and GE81112 (GE)[10–13]. Ksg is an aminoglycoside antibiotic[14] that was defined as a translation initiation inhibitor based on its ability to inhibit binding of initiator tRNA to the P-site of mRNA-containing 30S subunits[15–17]. Structures of Ksg have been determined on *Thermus thermophilus* 30S at 3.35 Å[17], on the *Escherichia coli* 70S without tRNAs and mRNA at 3.5 Å[18], and more recently at a higher resolution of 2.04 Å[19]. Two binding sites were observed on the *T. thermophilus* 30S[17], one

[1]Institute for Biochemistry and Molecular Biology, University of Hamburg, 20146 Hamburg, Germany. [2]Laboratory of Biomolecules, Faculty of Health Sciences, Universidad Peruana de Ciencias Aplicadas (UPC), 15023 Lima, Peru. [3]Department of Cellular, Computational and Integrative Biology – CIBIO, University of Trento, 38122 Trento, Italy. [4]Laboratory of Genetics of Microorganisms and Microbial Biotechnology, School of Biosciences and Veterinary Medicine, University of Camerino, 62032 Camerino, MC, Italy. [5]These authors contributed equally: Haaris A. Safdari, Martino Morici, Ana Sanchez-Castro. ✉e-mail: pmilon@upc.pe; Daniel.Wilson@chemie.uni-hamburg.de

of which was common to the *E. coli* structures[18,19]. This common site is considered the primary binding site because it correlates with chemical protection sites and Ksg resistance mutations located at A794 and G926 of the 16S rRNA[20,21]. Based on this location, Ksg binds within the path of the mRNA and does not overlap with the binding position of the initiator tRNA, leading to the suggestion that its effect on initiator tRNA binding is indirect[17,18]. However, to date, there are no structures of Ksg bound to translation initiation complexes so the exact step of action remains to be determined.

Ede is a pentapeptide antibiotic that inhibits binding of initiator tRNA to the small subunit of both prokaryotes and eukaryotes, and is thus a universal inhibitor of translation initiation[11,22,23]. Structures of Ede have been determined on the vacant *T. thermophilus* 30S at 4.5 Å[24] and on the yeast 80S ribosome at 3.1 Å[25], revealing distinct binding sites on bacterial and eukaryotic ribosomes. On the yeast 80S, the Ede binding site overlaps that of the mRNA and not the initiator tRNA[25], whereas, on the *T. thermophilus* 30S, Ede overlaps the binding position of the mRNA and the initiator tRNA, suggesting that the inhibition of initiator tRNA binding to the 30S subunit[22] is due to a direct steric clash[24]. However, structures of Ede within bacterial initiation complexes are lacking, which will be necessary to reveal the physiological binding site of Ede during translation initiation and elucidate its mechanism of action.

GE is a tetrapeptide antibiotic that interferes with binding of the initiator tRNA to the 30S[26,27]. An X-ray structure of GE on the *T. thermophilus* 30S at 3.5 Å suggested that GE interacts with and distorts the anticodon stem-loop of the initiator tRNA[27], however, this is inconsistent with previous biochemistry indicating that GE strongly protects 16S rRNA nucleotide G693 in helix 23 of the 16S rRNA from chemical modification[26]. A subsequent structure of a 30S-PIC stalled by GE was determined, but the 13.5 Å-resolution was insufficient to visualize the drug[28]. Thus, a structure of GE in complex with a 30S initiation complex at higher resolution is required to provide structural insights into the binding site of GE on the 30S subunit and its mechanism of action.

Here we have determined cryo-electron microscopy (cryo-EM) structures of Ksg, Ede and GE within the context of *E. coli* 30S initiation intermediate complexes at resolutions of 2.0–2.9 Å. While Ksg is observed to bind in the E-site as previously reported[17,18], we observe distinct binding sites for Ede and GE compared to those reported previously by X-ray crystallography on *T. thermophilus* 30S[24,27]. Specifically, we observe that all three antibiotics bind in the E-site within the mRNA path and perturb the transition of the 30S-PIC into a 30S-IC. Ede and Ksg interfere with the stable binding of initiator tRNA and prevent start codon recognition, whereas GE stalls the 30S PIC, yet allows AUG recognition by initiator tRNA. Ultimately, we show that all three antibiotics inhibit translation initiation by interfering with 30S-IC assembly and, therefore, precluding 70S-IC formation and mRNA translation.

## Results

### Cryo-EM structures of Ksg-30S initiation complexes
To determine structures of 30S initiation complexes formed in the presence of Ksg, we incubated purified 30S subunits, IF1, IF2, IF3, fMet-tRNA^fMet and MF-mRNA (Met-Phe-mRNA) with Ksg and analysed the resulting Ksg-30S complexes using single particle cryo-EM. In silico sorting revealed one major population (359,652 particles, 76%) comprising 30S subunits with mRNA, IF1 and IF3, as well as a second minor population (61,790 particles, 13%) with the additional presence of initiator tRNA (Supplementary Fig. 1). These two subpopulations, with and without initiator tRNA, could be refined to average resolutions of 2.9 Å and 2.5 Å, respectively (Supplementary Fig. 2). In the Ksg-30S structure without initiator tRNA, the body of the 30S subunit is well-defined, with highly-resolved density for IF1 and IF3 (Fig. 1a). By contrast, the head of the 30S subunit is highly mobile, as evident from the poor quality of the density (Fig. 1a) and low local resolution of the head (Supplementary Fig. 2). Therefore, we performed focused refinement

on the 30S-body, which further improved the resolution (2.4 Å) and density quality for the 30S-body, IF1 and IF3 (Fig. 1b and Supplementary Fig. 2). In this cryo-EM map, there is extra density that we could unambiguously assign to Ksg (Fig. 1b, c). Ksg was bound at the tip of helix 23 (h23) of the 16S rRNA, adjacent to h28 and h44 (Fig. 1d), as observed previously on vacant 30S or 70S ribosomes[17–19]. We do not observe any additional binding sites for Ksg, suggesting that the secondary binding site observed previously may be specific for *T. thermophilus*[17]. The quality of the cryo-EM map enables a precise description of direct and indirect water-mediated interactions between Ksg and the 16S rRNA (Fig. 1e,f and Supplementary Movie 1). This binding mode, including water-mediated interactions, of Ksg with the *E. coli* 30S initiation complex determined here is consistent with that reported previously for Ksg on the vacant *E. coli* 70S ribosome at 2.0 Å[19], whereas there are slight differences compared to the previous structures determined at lower (3.35–3.5 Å) resolutions[17,18] (Supplementary Fig. 1a–d). The interactions of Ksg with the 16S rRNA are consistent with chemical protection of A794 and G926 by Ksg[21] as well as Ksg resistance arising from mutations at nucleotides A794 and G926 (ref. 29).

### Ksg alters the path of the mRNA through the A-, P- and E-sites
Previous studies have demonstrated that Ksg has only a minor effect on binding of mRNA to 30S subunits (and 70S ribosomes), even at concentrations of up to 1 mM[17]. Indeed, we observe density for the Shine-Dalgarno (SD)-antiSD helix that forms between the mRNA and the 3' end of the 16S rRNA, as well as additional density for the mRNA extending into A-, P- and E-sites (Fig. 1a), which is lost at higher resolution (Fig. 1b), presumably due to its flexibility. Although the additional density is partially fragmented, strong regions of density were observed adjacent to 16S rRNA nucleotides G693, A790, G926 and C1400, suggesting stacking interactions of mRNA nucleotides with these bases (Fig. 1g, Supplementary Fig. 1e). Therefore, we generated a tentative model for the path of the mRNA in the presence of Ksg (Fig. 1g), which is similar, but distinct, from the path reported in a previous structure of 30-IC without initiator tRNA (Fig. 1h and Supplementary Fig. 1f)[6]. Based on this comparison, it appears that the +1 and +3 positions (the A and G of the AUG start codon, respectively) would be stacked on A790 and C1400, respectively, whereas the -1 and -3 positions (E-site codon) appear to stack on G926 and G693, respectively (Fig. 1g and Supplementary Fig. 1e). Interestingly, the kasugamine tail of Ksg can form stacking interactions with the base in the -2 position of the E-site codon of the mRNA (Fig. 1g and Supplementary Fig. 1e). We also compared the position of Ksg with that of the mRNA and accommodated initiator tRNA from a previously reported 30S initiation complex formed in the absence of Ksg[6] (Fig. 1i and Supplementary Fig. 1g, h). In this case, Ksg sits directly in the path of the mRNA, generating a large steric clash with the second and third nucleotide (-1 and -2 positions) of the E-site codon of the mRNA (Fig. 1i), consistent with the proposal that Ksg interferes with initiator tRNA binding indirectly by perturbing the placement of the mRNA[17,18].

### Cryo-EM structures of Ede-30S initiation complexes
To determine a higher resolution structure of Ede on a bacterial 30S, but also within the context of a 30S initiation complex, we analysed Ede-30S initiation complexes using cryo-EM. In silico sorting revealed that, like the Ksg-30S sample, the Ede-30S sample also contains one major population (523,691 particles, 80%) comprising 30S subunits with mRNA, IF1 and IF3, and a second minor population (82,223 particles, 13%) with the additional presence of fMet-tRNA^fMet (Supplementary Fig. 4). These two subpopulations, with and without fMet-tRNA^fMet, were refined to average resolutions of 2.8 Å and 2.1 Å, respectively (Supplementary Fig. 5). As seen for the Ksg-30S structure without initiator tRNA (Fig. 1a), the 30S-body is well-defined for the Ede-30S structure without initiator tRNA, whereas the head is poorly

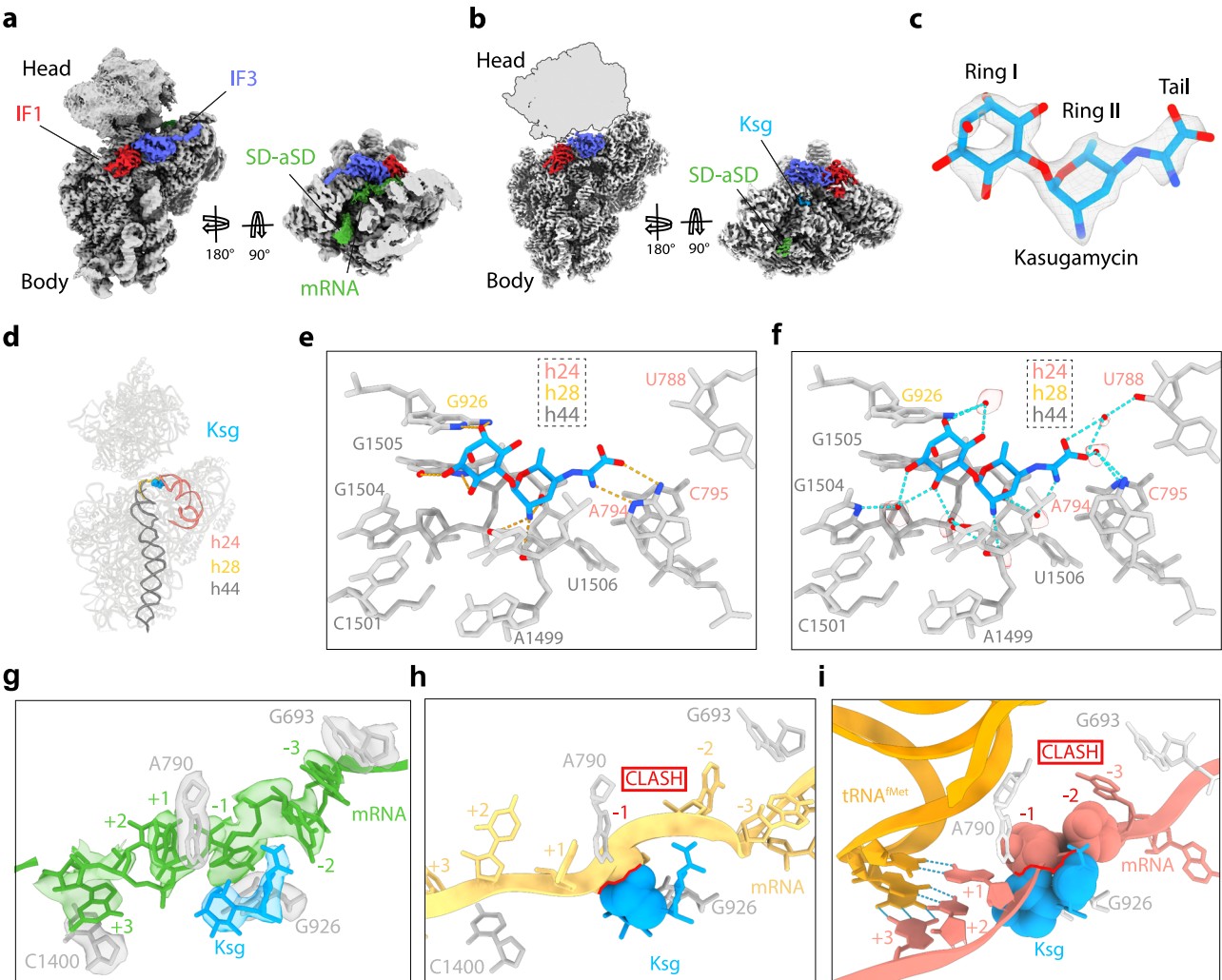

**Fig. 1 | Cryo-EM structure of Ksg-30S initiation intermediate complex. a** Refined map of Ksg-30S complex with IF1 (red) and IF3 (blue). Side view with transverse section showing the mRNA (green) with density for the Shine Dalgarno (SD)-antiSD (SD-aSD) helix and mRNA density extending into the A-, P- and E-sites. **b** Postprocessed map of Ksg-30S complex with Ksg (cyan), IF1 (red), IF3 (blue), and with the 30S head removed from map and shown as silhouette. **c** Density for Ksg shown from the postprocessed map with Ring I, Ring II and kasugamine tail labelled. **d** Helices of 16S rRNA that interact with Ksg include h24 (salmon), h28 (gold) and h44 (dark grey). **e** Putative hydrogen bond interactions (dashed lines)

between Ksg and the 16S rRNA nucleotides (grey). **f** Putative water-mediated interactions of Ksg with 30S subunit are shown as cyan dashed lines with cryo-EM density (grey mesh) shown for the modelled water molecules (red spheres). **g** Overlay of mRNA path (green) of Ksg (cyan) and 16S rRNA (grey) with density (transparent). mRNA positions are marked relative to P site codon (first position as +1). **h** Overlay of mRNA path (yellow) from 30S-PIC (PDB ID 5LMN)[6] showing a clash (red) with Ksg (cyan). **i** Overlay of mRNA path (salmon) and tRNA[fMet] (orange) from 30S-IC (PDB ID 5LMV)[6] showing a clash (red) with Ksg (cyan).

resolved (Fig. 2a and Supplementary Fig. 5). Focused refinement further improved the resolution (to 2.0 Å) and quality of the cryo-EM map density for the 30S-body, IF1 and IF3 (Fig. 2b and Supplementary Fig. 4). Although we observe density for the SD-antiSD helix in the Ede-30S maps, we do not observe any density for the mRNA extending into the A-, P- and E-sites (Fig. 2a, b). By contrast, there was extra density located in the E-site that we could unambiguously assign to Ede (Fig. 2b, c). In the Ede-30S complexes, Ede is bound at a location spanning the tips of h23 and h24 of the 16S rRNA (Fig. 2d). Interestingly, the binding position observed here on the *E. coli* 30S is distinct from that observed on the *T. thermophilus* 30S[24] (i.e. another bacterial ribosome), but very similar (if not identical within the resolution limits) to that observed for Ede bound to the yeast 80S (i.e. eukaryotic ribosome)[25] (Supplementary Fig. 6a-i). Additionally, the conformation of Ede on the *E. coli* 30S is also the same as observed in the yeast Ede-80S structure[25], such that the β-tyrosine and guanylspermidine tail come together to form a ring-like structure, whereas on the *T. thermophilus* 30S, the β-tyrosine and guanylspermidine ends of Ede are

located far apart and no ring-like structure is formed (Supplementary Fig. 6d-f).

## Conservation of the Ede binding site on the 30S subunit

The high resolution of the cryo-EM map of the Ede-30S complex enables a precise description of both direct and indirect water-mediated interactions (Fig. 2e, f and Supplementary Movie 2). Importantly, the β-tyrosine end of Ede interacts with nucleotides (A789-A792) in h24, with the central region predominantly contacting the backbone of nucleotides (U1498, G1505 and U1506) within h44, but also G926 in h28 and C795 in h24, whereas the glycine linker establishes many water-mediated interactions with G693 in h23 (Fig. 2e, f). By contrast, the guanylspermidine tail, which folds back towards the β-tyrosine end, does not appear to make any specific interactions with the 30S (Fig. 2e, f), presumably explaining why this moiety has weaker density (Fig. 2c). The binding mode of Ede is consistent with chemical probing assays where Ede protects G693 and C795 from chemical modification[21]. While we have determined the structure of congener B

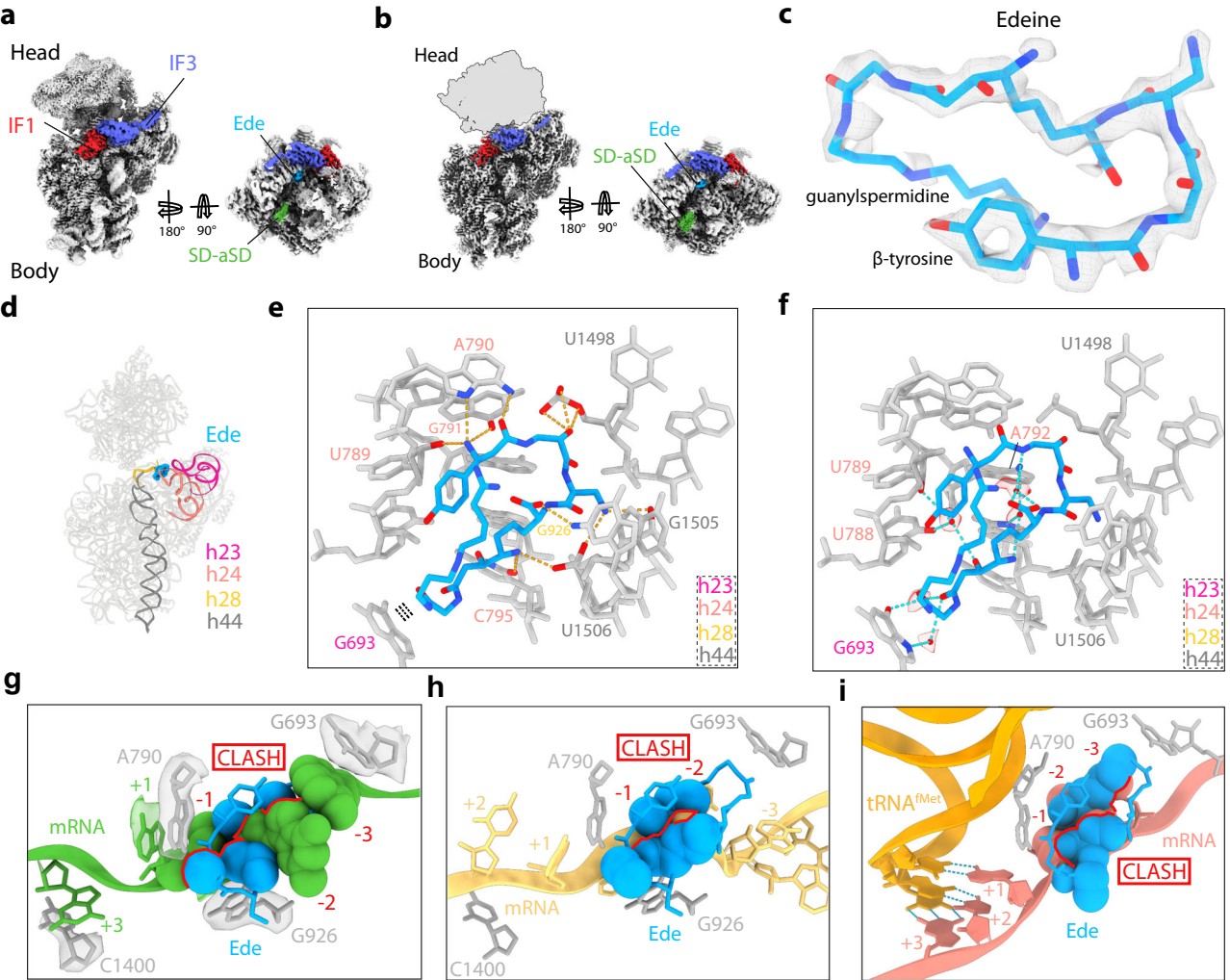

**Fig. 2 | Cryo-EM structure of Ede-30S initiation intermediate complex. a** Refined map of Ede-30S complex with IF1 (red) and IF3 (blue) with a transverse section showing the mRNA (green) with density for the Shine Dalgarno (SD)-antiSD (SD-aSD) helix. **b** Postprocessed cryo-EM map of Ede-30S complex with Ede (cyan), IF1 (red), IF3 (blue), and with the 30S head removed from map and shown as a silhouette. **c** Cryo-EM map density (transparent grey) for Ede (cyan) shown from postprocessed map. **d** Helices of 16S rRNA which interact with Ede include h23 (dark pink), h24 (salmon), h28 (gold) and h44 (dark grey). **e** Putative hydrogen bond interactions (dashed lines) between Ede and the 16S rRNA nucleotides (grey). **f** Putative water-mediated interactions of Ede with 30S subunit are shown as cyan dashed lines with cryo-EM density (grey mesh) shown for the modelled water molecules (red spheres). **g** Overlay of mRNA path (green) of Ede (cyan) and 16S rRNA (grey) with density (transparent). mRNA positions are marked relative to P site codon (first position as +1). **h** Overlay of mRNA path (yellow) from 30S-PIC (PDB ID 5LMN)[6] showing a clash (red) with Ede (cyan). **i** Overlay of mRNA path (salmon) and tRNA$^{fMet}$ (orange) from 30S-IC (PDB ID 5LMV)[6] showing a clash (red) with Ede (cyan).

of Ede on the 30S, generating in silico models for congeners A, C and D of Ede on the 30S revealed that the small differences in chemical structure are unlikely to influence their mode of interaction (Supplementary Fig. 7a-e). In fact, we suggest that the hydroxyl of the β-tyrosine and the tail of the guanylspermidine moiety of Ede would be excellent sites for introducing chemical moieties to develop improved derivatives. The binding position of Ede on the *T. thermophilus* 30S at 4.5 Å[24] would not lead to protection of G693 and C795 and, in fact, the structure does not appear to be fully refined since severe steric clashes exist between Ede and the 16S rRNA (Supplementary Fig. 8a, b). Similarly, superimposition of the binding position of Ede from the *T. thermophilus* 30S[24] with other bacterial ribosome structures also generates clashes (Supplementary Fig. 8c, d), leading us to suggest that the drug has been modelled incorrectly due to the low resolution and/or the binding site may be distorted due to crystal packing artefacts within the *T. thermophilus* 30S crystals. By contrast, alignment of the Ede binding site observed here (and in yeast 80S)[25] onto ribosomes from other species, including *T. thermophilus* and humans, illustrates

the high conservation of the binding site (Supplementary Fig. 8e-h), which is consistent with the ability of Ede to inhibit translation on both bacterial and eukaryotic ribosomes[11].

## Ede overlaps the mRNA path and indirectly inhibits initiator tRNA binding

The binding position of Ede on the *T. thermophilus* 30S led to the suggestion that Ede directly interferes with initiator tRNA binding due to the steric clash between the guanylspermidine moiety of Ede and the anticodon stem-loop of the P-site tRNA, whereas overlap with the mRNA is minimal[24] (Supplementary Fig. 6c). By contrast, we observe little density for the mRNA in the A-, P- and E-sites in the presence of Ede (Fig. 2a,b). To investigate the relative position of Ede and the mRNA, we aligned the mRNAs from our Ksg-30S complex (as seen in Fig. 1g), as well as mRNA from previous studies of *T. thermophilus* 30S initiation complexes with and without initiator tRNA (Fig. 2g-i)[6]. This clearly shows that the binding of Ede on the *E. coli* 30S creates a large blockage for each mRNA path, regardless of the absence or presence

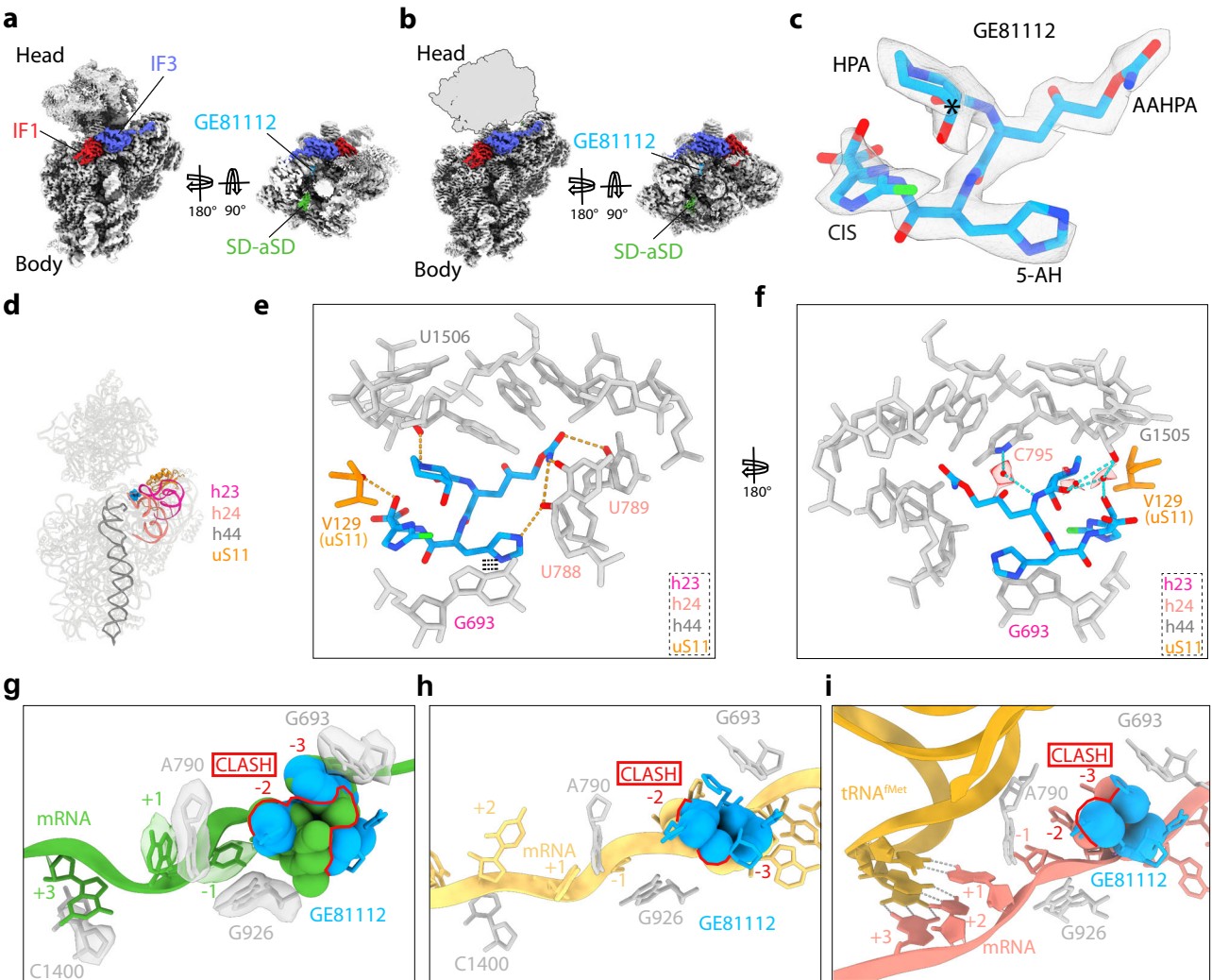

**Fig. 3 | Cryo-EM structure of the GE-30S initiation intermediate complex.**
**a** Refined map of GE-30S complex with IF1 (red) and IF3 (blue) with a transverse section showing the mRNA (green) with density for the Shine Dalgarno (SD)-antiSD (SD-aSD) helix. **b** Postprocessed cryo-EM map of GE-30S complex with GE (cyan), IF1 (red), IF3 (blue), and with the 30S head removed from map and shown as a silhouette. **c** Cryo-EM map density (transparent grey) for GE (cyan) shown from postprocessed map. An asterisk is shown on carbon with correct stereochemistry determined here. **d** Helices of 16S rRNA which interact with Ede include h24 (salmon), h44 (dark grey) and ribosomal protein uS11 (orange). **e** Putative hydrogen bond interactions (dashed lines) between GE and the 16S rRNA nucleotides (grey) and uS11 (orange). **f** Putative water-mediated interactions of GE with 30S subunit are shown as cyan dashed lines with cryo-EM density (grey mesh) shown for the modelled water molecules (red spheres). **g** Overlay of mRNA path (green) of GE (cyan) and 16S rRNA (grey) with density (transparent). mRNA positions are marked relative to the P site codon (first position as +1). **h** Overlay of mRNA path (yellow) from 30S-PIC (PDB ID 5LMN)[6] showing a clash (red) with GE (cyan). **i** Overlay of mRNA path (salmon) and tRNA[fMet] (orange) from 30S-IC (PDB ID 5LMV)[6] showing a clash (red) with GE (cyan).

of initiator tRNA. Specifically, the superimposition shows that Ede overlaps the position of the -1 and -2 nucleotides of the mRNA that are located in the E-site (Fig. 2g-i). Moreover, we observe no overlap between the binding position of Ede and that of the accommodated initiator tRNA at the P-site of the 30S (Fig. 2i)[6], leading us to conclude that the observed inhibition by Ede of P-tRNA binding[22] operates indirectly through perturbation of the path of the mRNA, rather than by directly and sterically blocking P-site tRNA binding, as proposed previously[24].

## Cryo-EM structures of GE-30S initiation complexes

To determine a higher resolution structure of GE within the context of a 30S initiation complex, we analysed GE-30S initiation complexes using cryo-EM. In silico sorting revealed that like the Ksg- and Ede-30S samples, the GE-30S sample also contains one major population (601,047 particles, 63%) comprising 30S subunits with mRNA, IF1 and IF3, and a second minor population (69,483 particles, 7%) with the

additional presence of initiator tRNA and IF2 (Supplementary Fig. 9). The main subpopulation without initiator tRNA could be further subsorted and then refined to yield a cryo-EM structure with an average resolution of 2.4 Å (Fig. 3 and Supplementary Fig. 9). Further sorting of the subpopulation with initiator tRNA revealed two distinct classes, with and without the additional presence of IF2 (Supplementary Fig. 9), which were refined to average resolutions of 2.9 Å and 2.8 Å, respectively (Supplementary Fig. 9,10). As seen for the Ksg- and Ede-30S-IF1-IF3 structures without initiator tRNA (Figs. 1a, 2a), the 30S-body is well-defined for the GE-30S structure, whereas the head appears highly flexible (Fig. 3a and Supplementary Fig. 10). Focused refinement further improved the resolution (to 2.3 Å) and quality of the cryo-EM map density for the 30S-body, IF1 and IF3 (Fig. 3b and Supplementary Fig. 10). Although we observe density for the SD-antiSD helix in the GE-30S maps, the density for the mRNA is not observed extending into the A-, P- and E-sites (Fig. 3a,b). However, there is an extra density located in the E-site that could be unambiguously assigned to GE (Fig. 3b,c).

The density was consistent with the revision of the stereochemistry for 3-hydroxypipecolic acid moiety of GE[30] (Fig. 3c), where one carbon atom had an inverse configuration leading to a change in sugar puckering as compared to that reported originally[12].

### Interaction of GE with the E-site of the 30S subunit

In the GE-30S complexes, the binding site for GE is located at the tips of h23 and h24 (Fig. 3d), which differs from that observed in the X-ray structure of GE bound to the *T. thermophilus* 30S[27] (Supplementary Fig. 11a-c). We observe that GE binds in the E-site where it makes extensive direct and water-mediated interactions with nucleotides of the 16S rRNA (Fig. 3e,f and Supplementary Movie 3). Specifically, the 3-hydroxypipecolic acid (HPA) moiety of GE contacts nucleotides in h44, including a hydrogen bond with the backbone of U1506 (Fig. 3e). The 2-amino-5-[(aminocarbonyl)oxy]-4-hydroxypentanoic acid (AAHPA) of GE interacts with U788 and U789 in h24, whereas the 5-amino-histidine (5AH) stacks upon G693 of h23 and can also hydrogen bond with U788 of h24 (Fig. 3e). The interaction of the 5AH of GE with G693 likely explains the protection of this base from modification by GE[26]. In addition, the 5-chloro-2-imidazolylserine (CIS) of GE can hydrogen bond with Val129 (Fig. 3e), which is the most C-terminal residue located in the long C-terminal tail of ribosomal protein uS11. Moreover, multiple water molecules mediate a network of indirect interactions between GE and the 16S rRNA (Fig. 3f). We note that at least three intramolecular hydrogen bonds can be formed between different moieties of GE (Supplementary Fig. 11d), which are likely to be important for adopting the conformation that is competent for ribosome binding. In particular, the 3-hydroxyl group of HPA can form hydrogen bonds with the nitrogen in the imidazole ring of the CIS moiety, as well as with the backbone nitrogen of 5AH (Supplementary Fig. 11d). These intramolecular interactions are likely to be critical for activity of GE since GE containing a *cis*-configuration of the HPA ring was completely inactive[30]. We predict that the *cis*-configuration of the HPA would not only disfavour the formation of the intramolecular interactions, but, on the ribosome, would extend towards and likely clash with U1506 (Supplementary Fig. 11e, f).

### A conserved binding site for GE on the 30S subunit

Modelling of the other congeners of GE, namely B and B1, onto the 30S, based on the structure of the 30S initiation complex with GE congener A determined here, reveals that the chemical substitutions that define the different congeners are not likely to be involved in forming interactions with the 30S. Thus, the binding site and interactions of GE81112A determined here are likely to be similar, if not identical, for the other GE congeners (Supplementary Fig. 12). Moreover, the binding site for GE on the 30S is highly conserved, not only between bacterial ribosomes of different species, such as *T. thermophilus*, but also on eukaryotic ribosomes, such as rabbit or human (Supplementary Fig. 13), consistent with the potent inhibitory activity observed for GE81112A in rabbit reticulocyte lysate-based in vitro translation assays[31]. Thus, while it is possible that GE81112B has a distinct binding site on *T. thermophilus* 30S as reported previously[27], we believe that this may have arisen due to crystal packing artefacts within the *T. thermophilus* 30S crystals. In the previous study[27], GE81112B was bound in the P-site of the 30S subunit, where it is proposed to cause a distortion in the anticodon-stem loop of the initiator tRNA[27]. However, in the structures determined here, with and without tRNA, GE is located in the E-site where it overlaps with the -1 and -2 nucleotides of the E-site codon of the mRNA from a previous 30S-IC complex[6] (Fig. 3g-i). Furthermore, unlike the previous study[27], we observe no interaction or overlap between GE and an accommodated initiator tRNA on the 30S (Fig. 3i)[6], leading us to conclude that the observed inhibition by GE of fMet-tRNA[fMet] binding[26] arises indirectly via perturbing the path of the mRNA through the E-site, rather than by directly interfering with P-site tRNA binding. We note that a recent independent study[32] visualized a similar binding site for GE81112 on the *E. coli* ribosome as observed here.

### Ksg, Ede and GE inhibit 30S-IC formation

The observation that the major subpopulations of the Ksg-, Ede- and GE-30S complexes lacked tRNA is consistent with the suggestion that the drugs interfere with initiator tRNA binding and/or accommodation in the P-site. Nevertheless, for all three antibiotics, we observe minor subpopulations (7-13%) of 30S complexes that contained initiator tRNA (Fig. 4a-c and Supplementary Figs. 1,4,9). For Ksg and Ede, the density for the head of the 30S subunit as well as the initiator tRNA are poorly ordered, indicating high flexibility (Supplementary Figs. 2, 5), which precluded models being generated. Based on the density, the overall conformation of the 30S head in these complexes appears to be open, with the initiator tRNA in an unaccommodated state, characterized by no defined anticodon-codon interaction with the mRNA (Fig. 4a, b and Supplementary Fig. 14). We, therefore, conclude that, although initiator tRNA can bind to a small population of Ksg-30S and Ede-30S complexes, the transition from the open to the closed conformation that occurs upon initiator tRNA accommodation is disfavored. Presumably, this is because both Ksg and Ede sterically occlude the path of the mRNA in the E-site (Figs. 1i and 2i), and thereby perturb the necessary placement of the start codon in the P-site that is required for codon-anticodon interaction with the initiator tRNA.

In contrast to the Ksg- and Ede-30S complexes, the density for the head and initiator tRNA in the GE-30S complex is better resolved, enabling models for the head and anticodon-stem of the initiator tRNA to be built and refined (Fig. 4c, Supplementary Movie 4 and Supplementary Table 1). In the GE-30S complex, the head adopts a closed conformation and the initiator tRNA is accommodated (Supplementary Fig. 14i-l), such that the anticodon of the initiator tRNA base pairs with the AUG start codon of the mRNA in the P-site of the 30S subunit (Supplementary Movie 4). Thus, the GE-30S complex has transitioned from a 30S-PIC towards a state resembling a 30S-IC. However, we do not observe the subsequent shift of the CTD of IF3 that is required for the 50S to join and form a 70S-IC[6]. No clear differences are observed in the GE-30S complexes with or without IF2 (Supplementary Fig. 9). Interestingly, although the initiator tRNA can base pair with the start codon, the presence of GE blocks the canonical path of the mRNA in the closed form (Fig. 4d) and prevents the initiator tRNA from displacing the CTD of IF3 from the 30S (Fig. 4e)[8]. Therefore, 50S joining cannot proceed because the clash between the IF3-CTD and 23S rRNA H69 remains, thereby preventing the association of the 50S (Fig. 4f).

To directly investigate whether GE, as well as Ksg and Ede, compromises 70S-IC formation, we assembled 30S initiation complexes in the absence and presence of each antibiotic and, upon addition of 50S, 70S-IC formation was monitored using light scattering in a stopped-flow instrument[1,33] (Fig. 4g). In the absence of the drugs, addition of 50S led to a rapid increase in light scattering over time, indicating 70S-IC formation. The time traces were biphasic consistent with a two-step mechanism of 50S binding to 30S ICs[1,9]. The presence of Ksg and Ede interfered with 50S joining (Fig. 4g) by reducing the efficiency over 5-fold, as seen from the amplitude, and the apparent averaged rates by 4- and 18-fold, respectively (Supplementary Table 2). This is consistent with their ability to lock translation initiation in an early 30S-PIC state, likely, with the open head and unaccommodated initiator tRNA (Fig. 4a,b). Similarly, GE also decreased 50S joining kinetics by 3-fold; however, 70S formation efficiency, as judged by the amplitude of the scattered light, was affected only by a factor of two (Fig. 4g, Supplementary Table 2). GE presumably allows the 30S to adopt a more 30S-IC-like state, consistent with the compound allowing start codon recognition, yet, reducing the kinetics of IF3 displacement as shown in the structures above (Fig. 3). All three compounds share a similar binding site as that of IF3; thus, to study their direct effect on the 30S-

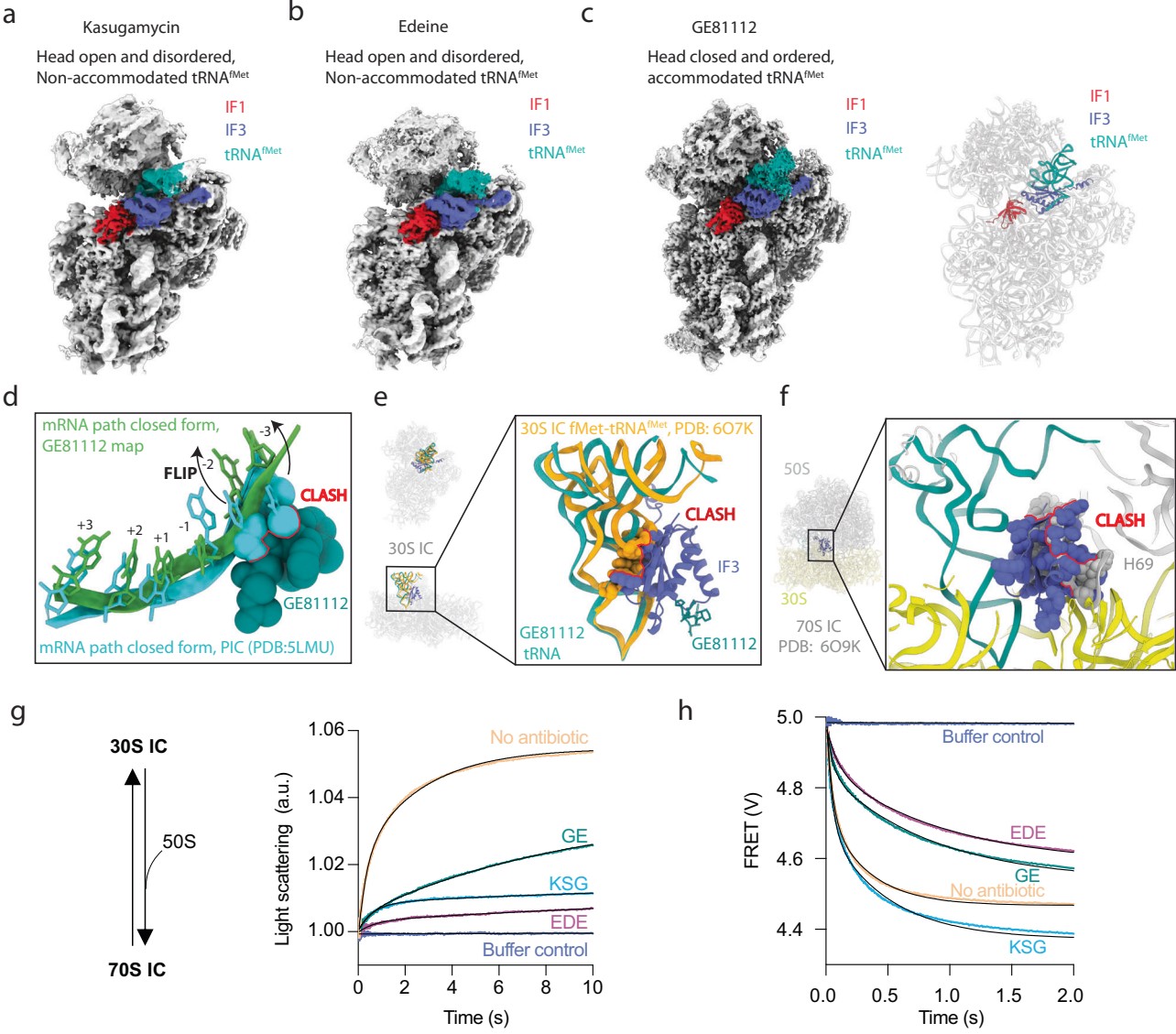

**Fig. 4 | Ksg, Ede, GE stabilize 30S initiation intermediates and inhibit of 70S-IC formation.** Refined maps of 30S-IF1-IF3-mRNA-tRNA complexes, with IF1 (red), IF3 (blue), fMet-tRNA^fMet (light sea green) labelled bound to Ksg (**a**), Ede (**b**), and GE (**c**). **d** Overlay of mRNA path of GE-30S-IF1-IF3-mRNA-tRNA complex (green) with mRNA path of the 30S-PIC (closed form; turquoise; PDB ID 5LMU)[6]. A clash (red) of GE with mRNA path shows the flip of bases at -2 and -3 positions. **e** Overlay of tRNA and IF3 from our GE-30S-IF1-IF3-mRNA-tRNA complex with tRNA from 30S-IC (PDB ID

6O7K)[7]. The fMet-tRNA^fMet of 30S-IC clashes with IF3 of our structure. **f** Alignment of GE-30S-IF1-IF3-mRNA-tRNA complex with 70S-IC (PDB ID 6O9K)[7]. IF3 of our structure clashes with H69 of 23S rRNA of 70S IC. **g** 70S IC formation by 50S binding to 30S ICs, as measured by light scattering. **h** IF3_DL dissociation measured by FRET upon mixing 30S-IF3_DL complexes with unlabeled IF3 using a stopped-flow apparatus. Buffer control traces indicate signal controls in the absence of the 50S (**h**) or unlabeled IF3 (**g**). Source data are provided as a Source Data file.

bound IF3, we performed experiments monitoring the exchange of IF3 on the 30S subunit. We used a double-labelled derivative of IF3 (IF3_DL) to monitor factor dissociation by intramolecular FRET[8]. IF3_DL binding to the 30S results in donor fluorescence increase, while the opposite is valid for factor dissociation. We observe that GE and Ede stabilize IF3 on the 30S as observed by a 2- and 4-fold reduction of the factor exchange rate ($k_{off}$), respectively (Supplementary Table 3). On the other hand, Ksg, albeit slightly increasing the signal amplitude, does not perturb the IF3 dissociation rate constant from the ternary 30S-IF3_DL-Ksg complex, suggesting that Ksg, differently from GE and Ede, does not directly affect the early 30S-IF3 complex (Fig. 4h, Supplementary Table 3).

## Discussion

Here we have determined structures of 30S initiation intermediate complexes bound with Ksg, Ede and GE, revealing that all three

antibiotics bind at the E-site in a position that overlaps with the mRNA (Fig. 5a). This contrasts with previous studies reporting that Ede and GE bind at the P-site, such that the drugs directly interfere with the positioning of the initiator tRNA in the P-site[24,27]. Instead, we propose a general model for the mechanism of inhibition of Ksg, Ede and GE during translation initiation, identifying two distinct points of action for each inhibitor (Fig. 5b-h). Consistent with previous biochemical studies, we observe that Ksg, Ede and GE, do not strongly interfere with the binding of mRNA to the 30S (Fig. 5b, c). Similarly, binding of IF1 and IF3 is also not prevented by the presence of the drugs, but rather we show that the drugs have a stabilizing effect on IF3 exchange (Fig. 4h). Instead, we observe that in the presence of Ksg, Ede or GE, the majority of 30S lack the presence of initiator tRNA, which agrees with previous biochemical studies reporting that these drugs prevent binding of initiator tRNA to the 30S[17,18,22,26,27]. Consistently, we have formed 30S initiation complexes using the same components, but in the absence of

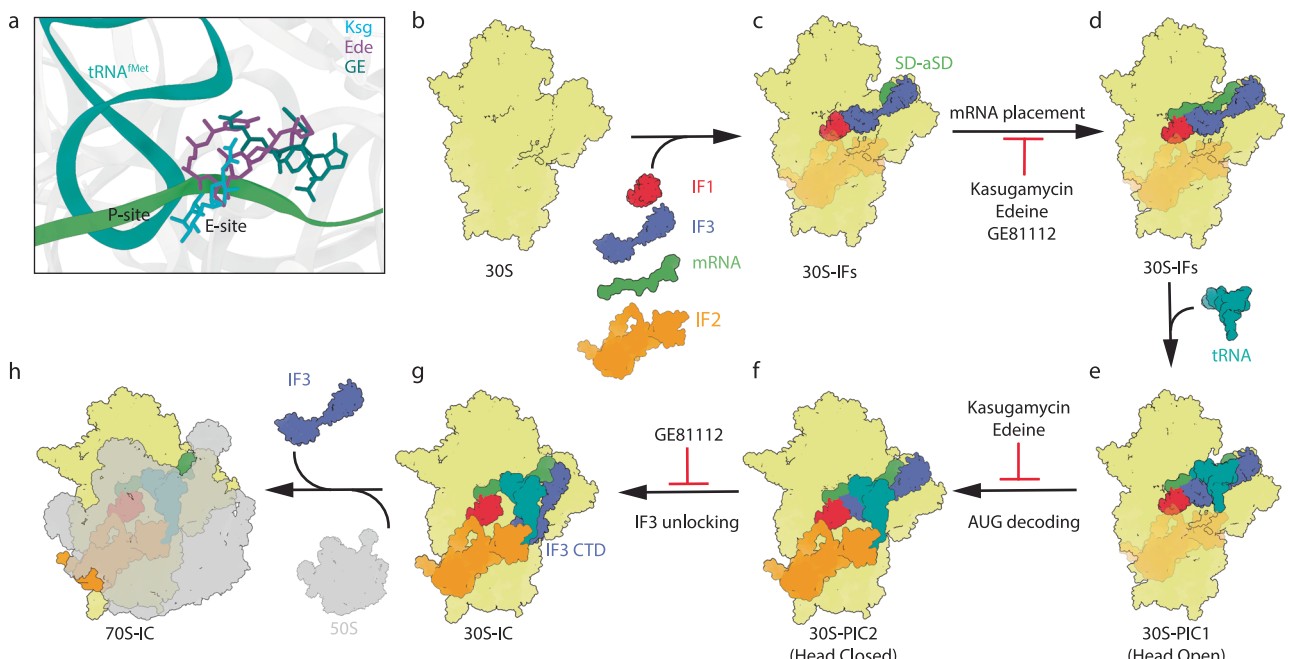

**Fig. 5 | Model for the mechanism of inhibition of Ksg, Ede and GE. a** Overall location of three antibiotics relative to the mRNA (green) with the initiator tRNA (green) in the P-site shown for reference. **b** 30S is joined by IF1, IF2, IF3 and mRNA to form 30S-IFs complex. **c** Ksg, Ede, and GE interfere with the mRNA placement across the mRNA path, though it allows the SD-aSD interaction. **d** tRNA joins the 30S-IFs complex to form 30S-PIC1 complex in which tRNA is non-accommodated (with 30S head open). **e** Conformation of tRNA with simultaneous head movement to form 30S-PIC2 complex with head closed and ordered (and tRNA accommodated). This transition from 30S-PIC1 to 30S-PIC2 is also inhibited by Ksg and Ede. **f** IF3-CTD moves away from the P site and forms 30S initiation complex (30S IC). GE81112 prevents IF3-CTD from leaving P site, hindering the formation of the 30S-IC. **g** 50S subunit joins the 30S-IC to form the 70S-IC. **h** 70S-IC formed to ready to be matured into 70S elongation-competent complex (70S-EC).

antibiotic, and observe that 90% of the 30S initiation complex contain initiator tRNA (Supplementary Fig. 15). Thus, we favor a model whereby Ksg, Ede and GE disrupt the path of the mRNA through the E-site (Fig. 5c, d), which in turn indirectly interferes with accommodation of the initiator tRNA in the P-site (Fig. 5d, e). However, we note that despite the presence of the drugs, binding of the initiator tRNA to the P-site is still possible since we observe a small population (7–13%) of drug-bound 30S complexes where initiator tRNA is also present (Fig. 4a-c). In the case of Ksg and Ede, these 30S-complexes are in an open conformation with an unaccommodated initiator tRNA, suggesting that Ksg and Ede interfere with the transition of the 30S head from an open to closed conformation, which occurs upon accommodation of the initiator tRNA at the P-site (Fig. 5e-f). By contrast, GE appears to allow the transition and, unlike Ksg and Ede, actually stabilizes a closed conformation of the head where the initiator tRNA is accommodated at the P-site, i.e. forming codon-anticodon interactions with the start codon of the mRNA (Fig. 5f). We could show that GE stabilizes IF3 on the 30S (Fig. 4h) and we observe all GE-30S complexes with the CTD of IF3 at the P-site position[6], allowing us to propose that GE interferes with IF3 release (Fig. 5f-g) during 50S joining and 70S-IC formation (Fig. 5g-h). Indeed, we could demonstrate using light scattering that GE, as well as Ksg and Ede, drastically reduce both, the kinetics and efficiency, of 70S-IC formation (Fig. 4g). Recent studies have shown that Ksg can also interfere with translation after 70S formation, and that the extent of inhibition is modulated by the nature of the mRNA, specifically, the nucleotide directly upstream of the start codon[34]. It will be interesting in the future to investigate whether similar effects are also observed for Ede and GE, and to provide a structural basis for the context-specificity.

In conclusion, we demonstrate that Ksg, Ede and GE have similar, but distinct binding sites within the E-site of the 30S subunit (Fig. 5a). On the one hand, the similarity in the binding site of the three antibiotics is reflected by their related mechanism of action to interfere with the stable binding of the initiator tRNA to the P-site (Fig. 5c,d). On the other hand, the difference in scaffolds and binding modes between the antibiotics leads to differences in the extent of overlap with the mRNA in the E-site, which in turn is likely to be reflected by the differences in the mechanism of action to stabilize the open or closed conformation of the 30S (Fig. 5e-g). Finally, we envisage that the detailed description of the binding sites, including the involvement of water-mediated interactions of Ksg, Ede and GE with the components of the 30S subunit reported here, will provide a basis for design of further improved inhibitors. In this regard, we note the binding positions of Ksg and GE (Fig. 5a) lend themselves to the generation of hybrid compounds formed by linking these two drugs.

## Methods

### Purification of ribosomes, mRNA, tRNA and initiation factors

30S and 50S ribosomal subunits were purified as detailed in ref. 35 by sucrose gradient centrifugation. Prior to use, the 30S subunits were re-activated with 21 mM $MgCl_2$ for 30 min at 37 °C. Raising the magnesium concentration (21 mM $MgCl_2$) restores essential ionic interactions and re-activates purified 30S subunits, as previously demonstrated by classical and more recent structural studies in ribosome biochemistry[36–38]. The MF-mRNA sequence (AAA CAA UUG GAG GAA UAA GGU aug UUU GGC GGA AAA CGA G) was synthesized and commercially obtained from Microsynth (Balgach, Switzerland). The fMet-tRNA^fMet was aminoacylated, formylated, and purified by HPLC following an adaptation from ref. 35. GTP was obtained commercially (Jena Bioscience, Germany). Wild-type initiation factors IF1, IF2, and IF3, as the IF3 E166C (for IF3_DL labelling), were obtained recombinantly following Ref. 8. Additionally, IF3 E166C was double- labeled (IF3_DL) using Atto 540Q maleimide fluorophore (Atto-Tec GmbH) in the C-terminal domain and Alexa 488-C5 maleimide fluorophore (Invitrogen, Thermo Fisher) in the N-terminal domain following the method previously mentioned[39]. All the proteins had a purity > 98%.

### Pre-steady state measurements and analysis FRET experiments

Before use, all materials were centrifuged at 10,000 × g at 10 °C for 10 min. All reactions were performed in $TAKM_7$ buffer (20 mM Tris-HCl pH 7.4, 30 mM KCl, 70 mM $NH_4Ac$, 7 mM $MgCl_2$) mixed with 100 µM GTP. For FRET assays, the following final concentrations were used unless stated otherwise: 0.05 µM of activated 30S subunit, 0.05 µM $IF3_{DL}$, 0.15 µM IF1 and IF2, 0.25 µM MF-mRNA and 0.15 µM fMet-$tRNA^{fMet}$, and 0.15 µM 50S. After measuring antibiotics $K_d$, we established the reaction concentrations for antibiotics to 5 µM for GE, and Ede, and 100 µM for Ksg. 30S initiation mixtures were prepared with the antibiotics before loading the stopped-flow apparatus as previously described[40]. In short, both fluorescence and light scattering assays were performed using a SX-20 stopped-flow apparatus (Applied Photophysics) by mixing equal volumes of two mixtures and recording after an external trigger. FRET was recorded by measuring donor fluorescence (Alexa 488 C5 fluorophore) with the excitation wavelength set to 470 nm and after passing a 515 nm optical cut-off filter. For light scattering assays, to observe the formation of 70S-IC complexes, the monochromator was set at 430 nm[41] and scattered light was measured without any optical filter. All reactions were measured at 25 °C. A thousand points were obtained logarithmically, measuring 5−7 replicates. The recorded time was set accordingly to the different reactions until a plateau was observed. The obtained data was averaged and plotted in GraphPad Prism 10 program (GraphPad Software, version 10.3.0). For non-linear regression fitting, a one- or two-step exponential equations were applied.

### Preparation of cryo-EM antibiotic-ribosome complexes

Complexes were prepared as above with the following adjustments. Mix1 containing 0.15 µM reactivated 30S, 100 µM GTP, 0.45 µM of IF3, IF2 and, IF1, and 0.75 µM fMet-$tRNA^{fMet}$ was incubated for 10 mins at 37 °C. Individual reactions were prepared by mixing in the following order: Mix1, 200 µM antibiotic Ksg, or 100 µM edeine B or 100 µM GE81112A, and 0.75 µM mRNA. Oversaturating antibiotic concentrations were used to ensure maximal occupancy on the 30S for their primary and eventually, secondary binding sites. The mixtures were briefly centrifuged and incubated for 20 mins at 37 °C. These mixtures were then centrifuged at maximum speed in a microfuge for 5 mins at 20 °C. A last incubation for 10 mins at 20-25 °C was performed before applying 3.5 µL of the sample to the cryo-EM grids.

### Preparation of cryo-EM grids

3.5 µL of the antibiotic-30S complexes were applied to grids (Quantifoil, Cu, 300 mesh, R3/3 with 3 nm carbon, Product: C3-C18nCu30-01) which had been freshly glow-discharged using a GloQube® Plus (Quorum Technologies) in negative charge at 25 mA for 40 sec. Sample vitrification was performed using a mixture of ethane/propane in 1:2 ratio in a Vitrobot Mark IV (ThermoScientific), with the chamber set to 100% relative humidity and 4 °C, and blotting performed for 3.5 sec with -1 blot force with Whatman 597 blotting paper. The grids were then clipped into autogrid cartridges and stored in liquid nitrogen until data collection.

### Data acquisition

All three cryo-EM datasets were collected using a Titan Krios G3i (Thermo Fisher Scientific/FEI) transmission electron microscope equipped with a K3 direct electron detector, post column GIF (energy filter) and Fringe-Free Imaging (FFI) setup at the Center for Structural Systems Biology (CSSB), Hamburg. GIF fine-centering was performed, and the K3 gain references were acquired prior to data collection. Data collection was performed using EPU (version 3.2.0.4775REL). Movies were recorded at defocus values from −0.4 µm to −1.2 µm with step size of 0.2 between holes at a magnification of 105,000×, which corresponds to the pixel size of 0.832 Å per pixel at the specimen level (super-resolution 0.416 Å per pixel) binned twice on the fly through EPU for all the datasets. During the 1.91 sec exposure in nanoprobe mode, 35 frames (1.14 e⁻ per frame per Å²) were collected with a total dose of around 40 e⁻ per Å². (15 e/px/s over an empty area on the camera level). C2 aperture of 70 µm was inserted with beam spot size of 7. BioQuantum energy filter set to 20 eV cut-off was used to remove inelastically scattered electrons. Final objective astigmatism correction <1 nm and auto coma-free alignment <50 nm was achieved using AutoCTF function of Sherpa (version 2.11.1). A total 9,576 micrographs for Ksg-30S complex; 8,360 micrographs for Ede-30S complex and 9736 micrographs were collected for GE81112A-30S complex, and saved as tiff gain corrected files.

### Cryo-EM data processing

RELION v4.0.1[42,43] was used for image processing, unless otherwise specified. For motion correction, RELION's implementation of MotionCor2 with 7×5 patches[44], and, for initial contrast transfer function (CTF) estimation, CTFFIND version 4.1.14[45], were employed. Particle picking was done using crYOLO[46] and imported to RELION. After 2D classification, all ribosome like particles were selected, extracted with pixel size of 1.664 Å, and 60 Å low pass filtered 30S ribosome (PDB ID 7OE1) was used as reference to perform 3D consensus refinement of these particles. With this 3D refined map, 3D classification was performed without angular sampling. All classes that contained 30S subunits at high resolution were used for further processing. Particles with homogenous 3D class distribution were re-extracted using smaller pixel size and subjected to 3D refinements. Subsequently, CTF refinements were performed to correct for anisotropic magnification, defocus and astigmatism, beam tilt, trefoil and higher order aberration followed by Bayesian polishing[47]. For partial signal subtraction, masks around the region of interest were created. Masking of 3D maps was done using soft mask to avoid artificial correlation and extended to several pixels to avoid overlap with volume.

For Ksg-30S dataset, after motion correction and CTF estimation, 522,384 particles were picked using crYOLO[46] (Supplementary Fig. 1). 2D classification with 100 classes was performed and 471,869 ribosome-like particles were selected for further processing. These particles were used as input for consensus refinement against 30S map and then used as input for 3D classification (without angular sampling) with 5 classes. Three classes with high resolution features with different 30S head rotation were obtained; H1 (Head rotation state 1) with 64,742 particles; H2 with 332,938 particles and H3 with 23,672 particles and all displayed density for Ksg, IF1 and IF3. The fourth class with low resolution containing 12,345 particles and fifth class with h44 absent (38,082 particles) were discarded. The first three classes were combined, resulting in 421,422 particles, and subjected to partial signal subtraction with a mask around IF3 and P-tRNA site enabling focused 3D classification with three output classes. One of the resulting classes had density for IF1, IF3 and fMet-$tRNA^{fMet}$ (61,790 particles), whereas the other two had density for only IF1 and IF3, but no tRNA. These two major classes without tRNA were combined (359,652 particles) and processed further. In particular, the resulting classes were reverted to original non-subtracted images, 3D refined, and CTF refined followed by Bayesian polishing[47]. For the major state of Ksg-30S complex containing IF1 and IF3, a final average resolution (gold-standard $FSC_{0.143}$) of 2.4 Å was obtained for masked reconstruction. Since the head of this structure was flexible, the body was masked and used for partial signal subtraction to remove the head from the final reconstruction. For the minor state of Ksg-30S complex with IF1, IF3 and fMet-$tRNA^{fMet}$ complex, a final average resolution (gold-standard $FSC_{0.143}$) of 2.9 Å for masked reconstruction was obtained (Supplementary Fig. 2).

For the Ede-30S dataset, after motion correction and CTF estimation, 870,707 particles were picked using crYOLO[46] (Supplementary Fig. 4). 2D classification with 100 classes was performed and 652,810 ribosome-like particles were selected for further processing. These particles were used as input for consensus refinement against a 30S

map and then used as input for 3D classification (without angular sampling) with 5 classes. Four classes with high resolution like features with different 30S head rotation were obtained; H1 (Head rotation state 1) with 154,508 particles; H2 with 76,765 particles, H3 with 276,575 particles and H4 with 98,066 particles and all of them had apparent density for Ede, IF1 and IF3. The fifth class lacked h44 (46,896 particles) and was therefore discarded. The first four classes with density for antibiotic, IF1 and IF3 were combined, resulting in 605,914 particles and subjected to partial signal subtraction with a mask around IF3 and P-tRNA site, enabling focused 3D classification to be performed with three output classes. One of these classes had density for IF1, IF3 and fMet-tRNA$^{fMet}$ (82,223 particles) and two others had density for only IF1 and IF3, but no tRNA. These latter two major classes were combined (523,691 particles) and processed further. In particular, the resulting classes were reverted to original non-subtracted images, 3D and CTF refined, followed by Bayesian polishing[47]. For the major state of Ede-30S complex with IF1 and IF3, a final average resolution (gold-standard FSC$_{0.143}$) of 2.1 Å for the masked reconstruction was obtained (Supplementary Fig. 5). Since the head of this map was flexible, a mask was applied to the body for partial signal subtraction to remove the head from the final reconstruction. For the minor state of Ede-30S complex with IF1, IF3 and fMet-tRNA$^{fMet}$, a final average resolution (gold-standard FSC$_{0.143}$) of 2.8 Å for the masked reconstruction was obtained (Supplementary Fig. 5).

For the GE-30S dataset, after motion correction and CTF estimation, 1,041,536 particles were picked using crYOLO[46]. 2D classification with 100 classes was performed and 953,815 ribosome-like particles were selected for further processing. These particles were used as input for consensus refinement against a 30S map and then used as an input for 3D classification (without angular sampling) with six output classes. Four classes with high resolution features of IF1, IF3 and different degrees of 30S head rotation, as well as IF2 and P-tRNA occupancies, were obtained. These classes were combined, resulting in 924,516 particles, which were subjected to partial signal subtraction with a mask around P-tRNA site (which also included signal from IF1, IF3 and surrounding 30S) and then focused 3D classification performed. After extensive sorting to achieve homogeneously sorted particle subpopulations, a total of six classes were obtained.

One of these classes was characterized by low resolution features and was therefore discarded (175,675 particles). A second class had clear density for IF1 and IF3 but no signal for P-tRNA (IF1-IF3 complex, 601,047 particles). Two other classes displayed density for IF1, IF3 and P-tRNA. In one of them (IF1-IF3-tRNAi tilted complex, 35,699 particles), the signal for the tRNA was very weak and was showing an abnormal placement on the 30S, while the second class displayed clear orientation bias (IF1-IF3-tRNAi biased complex, 17,150 particles). These two minor classes were discarded. A fourth class had high resolution features, stoichiometric IF1, IF3 and P-tRNA (IF1-IF3-tRNAi complex, 69,483 particles). In the fifth class, we observed IF1, IF3 and P-tRNA, however, the overall signal was noisy, therefore these particles were also discarded (IF1-IF3-tRNAi noisy complex, 25,462).

The second class (IF1-IF3 complex, 601,047 particles) was reverted to original non-subtracted images (30S-IF1-IF3-mRNA-GE81112A complex) to undergo further 3D classification (subsorting), yielding three classes. A first class clustered with low resolution particles (78,163 particles) and was therefore not processed further. The second class showed high resolution features and possessed both stoichiometric IF1 and IF3 (30S-IF1-IF3-mRNA-GE81112A complex, 457,875 particles) and was thus selected for further processing (see later). The third class showed no IFs and lacked density for h44 (30S-mRNA-GE81112A, no h44 complex, 65,009 particles), so it was also not processed further. The IF1-IF3-tRNAi complex (69,483 particles) class from the previous focus sorting was also reverted to original non-subtracted images to undergo 3D reconstruction, revealing the presence of density that could be attributed to IF2 (30S-IF1-sub-IF2-IF3-mRNA-tRNAi-GE81112A complex). Since IF2 was substoichiometric, partial signal subtraction was performed on these particles with mask around IF2 and focussed 3D classification was undertaken, yielding three classes. Two major classes contained IF1, IF3, P-tRNA and GE81112A and either lacked IF2 (33,966 particles) or contained stoichiometric IF2 (35,376 particles), whereas the third class was low resolution with very few particles (141 particles) and was therefore discarded.

The 30S-IF1-IF3-mRNA-GE81112A class (457,875 particles), 30S-IF1-IF3-mRNA-tRNAi-GE81112A class (33,966 particles) and the 30S-IF1-IF2-IF3-mRNA-tRNAi-GE81112A complex (35,376 particles) were CTF refined followed by Bayesian polishing[47]. After a second round of CTF refinement, the final average resolution (gold-standard FSC$_{0.143}$) was 2.3 Å, 2.8 Å and 2.9 Å, respectively, for masked reconstruction was obtained.

## Generation of molecular models

The molecular models of the 30S ribosomal subunits were based on the *E. coli* 30S ribosome subunit body (PDB ID 8CEP)[19]. Molecular models for edeine and GE81112 were generated initially using the final refined molecular model of kasugamycin dataset. Starting models with individual chains of ribosomal proteins and rRNA were rigid body fitted using ChimeraX[48] and modelled using Coot 0.9.8.92[49,50] from the CCP4 software suite version 8.0[51]. *E. coli* IF1 and IF3 models were generated using AlphaFold[52,53]. The mRNA model for kasugamycin was generated using PDB ID 5LMN[6] as a template to guide the position of nucleotides in the +1 and -1 positions. The initial model for the head of 30S subunit for 30S-IF1-IF3-mRNA-tRNAi-GE81112A class was based on PDB ID 8CA7[19]. The mRNA for 30S-GE81112-IF1-IF3-tRNA model was modelled de novo utilizing the codon-anticodon interaction. Initial fMet-tRNA$^{fMet}$ model was taken from PDB ID 6XZ7[54]. Model refinement was done using Servalcat[55]. Water, magnesium and potassium ions were designated according to model PDB ID 8CEP[19] and initially kept in a separate chain. Chain refine was used to place the water, magnesium and potassium ions into respective density and validated by difference map generated from servalcat refinement[55]. For antibiotic GE81112A, without available 3D structure, models were generated using ChemDraw (PerkinElmer Informatics) with structural restrains generated using aceDRG[56]. Manual adjustments using real space refinement function was done using Coot[49,50]. The final molecular models were validated using Phenix comprehensive cryo-EM validation tool in Phenix 1.20–4487[57]. The Molprobity server[58] was used to calculate map vs model cross correlation at Fourier Shell Correlation (FSC$_{0.5}$) for all maps. Angular distribution plot was made modifying the output of angdist tool deposited on Zenodo/Github (Ref: 10.5281/zenodo.4104053). For generating the model of *cis* configuration of the HPA ring in GE81112, the SMILES string of modified residue was generated using MarvinSketch with structural restrains generated using aceDRG[56].

## Figure preparation

UCSF ChimeraX v1.8[48] was used to isolate densities, color zone maps and visualize density images. Models were aligned using PyMol version 3.0 (Schrödinger). Figures were assembled with Adobe Illustrator v28.5.

## Reporting summary

Further information on research design is available in the Nature Portfolio Reporting Summary linked to this article.

## Data availability

Initial models for structure were generated based on the E. coli 30S subunit PDB ID 8CEP and PDB ID 8CA7 19. The cryo-electron microscopy maps for the antibiotic-ribosome complexes have been deposited in the EMDataBank with the accession code EMD-50320 (Ksg-PIC), EMD-51214 (Ksg-tRNA-PIC), EMD-50327 (Ede-PIC), EMD-51217 (Ede-tRNA-PIC), EMD-50476 (GE-PIC), EMD-50912 (GE-30S-

PIC2). The respective coordinates for electron-microscopy-based model of the antibiotic-ribosome complexes are deposited in the Protein Data Bank with the accession code PDB 9FCO (Ksg-PIC), PDB 9FDA (Ede-PIC), PDB 9FIB (GE-PIC) and PDB 9G06 (GE-30S-PIC2). Structures from prior studies were used in this work for comparison, alignments and modelling and are available in the Protein Data Bank, with PDB ID 8CEP, 7K00, 8CA7, 5LMN, 5LMQ, 5LMU, 5LMV, 6XZ7, 7OE1, 4V4H, 2HHH, 4U4N, 1I95, 4UG0, 5IWA. The cryo-EM map that was used as reference is available in the EM Data Bank, with EMD-12857. Source data are provided with this paper.

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

## Acknowledgements

This research was supported by the Deutsche Forschungsgemeinschaft (DFG, German Research Foundation) WI3285/12-1 (to D.N.W.) and the Concytec/Prociencia program grant PE501079419-2022 (to P.M.). Cryo-EM data collection was performed at the Multi-User CryoEM Facility at the Centre for Structural Systems Biology, Hamburg, supported by the Universität Hamburg and DFG grant numbers (INST 152/772-1|152/774-1|152/775-1|152/776-1|152/777-1 FUGG). The funders had no role in study design, data collection and analysis, decision to publish or preparation of the manuscript. We acknowledge financial support from the Open Access Publication Fund of Universität Hamburg.

## Author contributions

D.N.W. and P.M. designed the study. A.S.-C and A.M.G. isolated components for biochemical and structural studies. A.S.-C. optimized all functional complexes and performed biochemical validations. M.M. and H.A.S. prepared the cryo-EM samples. H.A.S. made grids and collected cryo-EM data. H.A.S. and M.M. processed the cryo-EM data, and built and refined the molecular models with help from H.P. A.D. and A.M.G. and A.F. helped with data interpretation. D.N.W. and P.M. wrote the paper with help from all authors.

## Funding

## Competing interests

The authors declare no competing interests.
