## [Transparent Peer Review file · Nature Communications]

The translation inhibitors kasugamycin, edeine and GE81112 target distinct steps during 30S initiation complex formation

Corresponding Author: Professor Daniel Wilson

Version 0:

Reviewer comments:

Reviewer #1

(Remarks to the Author)

The manuscript by Haaris et al., from the Wilson group, reports the results of a structural and functional investigation aiming at understanding the action mode of three known antibiotics that act on the 30S (small ribosomal subunit of the bacterial ribosome), Kasugamycin, Edeine and GE81112. The authors mainly present structures of the initiation complexes (including the full set of initiation factors reconstituted *in vitro*) bound to these antibiotics, one at a time. The structures reveal that all three antibiotics bind within the E-site of the 30S and disadvantage 30S initiation complex formation. It has been previously demonstrated that kasugamycin (Ksg) and edeine (Ede) affect early steps of 30S pre-initiation complex formation, while GE81112 (GE) stalls pre-initiation complex formation at a later step by impeding IF3 departure after the start codon recognition. However, until the current study, no structures of Ksg, Ede and/or GE bound to translation initiation complexes have been determined.

The determined cryo-EM structures of Ksg, Ede and GE within the context of *E. coli* 30S (intermediate) initiation complexes were solved at resolutions varying between 2.0 and 2.9 Å. The authors show that all three antibiotics inhibit translation initiation by interfering with 30S-IC assembly and therefore precluding 70S-IC formation and mRNA translation. The manuscript is well written and mostly pretty clear, with few exceptions. The figures are all required for the proper understanding of the manuscript and are, in addition, aesthetic. However, some effort could be made to streamline the figure a bit further (see reviewer's comments below), which remains totally optional and to the discretion of the authors. The reviewer only has few minor points:

Minor points:

In Results:

In "Cryo-EM structures of Ksg-30S initiation complexes"

The authors write "In silico sorting revealed one major population (359,652 particles, 76%) comprising 30S subunits with mRNA, IF1 and IF3, as well as a second minor population (61,790 particles, 13%) with the additional presence of initiator tRNA (Supplementary Fig. 1). These two subpopulations, with and without initiator tRNA, could be refined to average resolutions of 2.9 Å and 2.5 Å, respectively (Supplementary Fig. 2)." Figure S2 shows that the tRNA-dissociated class refined down to 2.4 and not 2.5 Å.

The authors write "Therefore, we performed focussed refinement on the 30S-body, which further improved the resolution (2.4 Å) and density quality for the 30S-body, IF1 and IF3 (Fig. 1b and Supplementary Fig. 2)."

The interest of showing the non-locally refined structure in panel 1a is very limited, I suggest its removal from the main figures.

The authors write "The quality of the cryo-EM map enabled a precise description of direct and indirect water-mediated interactions between Ksg and the 16S rRNA (Fig. 1e,f and Video S1)." I believe it's straightforward to show both kinds of interactions on the same panel, direct and indirect water mediated. therefore, I suggest the removal of one of the panels, the authors could keep panel f, for example, and add the direct interactions on the top.

In "Ksg alters the path of the mRNA through the A-, P- and E-sites"

The authors write "Although the additional density was partially fragmented, strong regions of density were observed adjacent to 16S rRNA nucleotides G693, A790, G926 and C1400, suggesting stacking interactions of mRNA nucleotides with these bases (Fig. 1g, ED Fig. 1e)."

The low resolution of the non-locally refined map makes difficult, and to some extent risky, to interpret these bits of mRNA densities, especially since their structure and precise orientation don't provide much of relevant insight beyond the presence of mRNA in its channel.

The authors write "Therefore, we generated a tentative model for the path of the mRNA in the presence of Ksg (Fig. 1g), which was similar, but distinct, from the path reported in a previous structure of 30S-IC lacking initiator tRNA (Fig. 1h and ED Fig. 1f)6"

Panel 1h is confusing as a main figure, as it could be seen as conveying the message that Ksg is incompatible with the binding of mRNA, which is the opposite of the message intended by the authors.

In "Cryo-EM structures of Ede-30S initiation complexes"

The authors write "As seen for the Ksg-30S structure without initiator tRNA (Fig. 1a), the 30S-body was well-defined for the Ede-30S structure without initiator tRNA, whereas the head was poorly resolved (Fig. 2a and Supplementary Fig. 4). Focussed refinement further improved the resolution (to 2.0 Å) and quality of the cryo-EM map density for the 30S-body, IF1 and IF3 (Fig. 2b and Supplementary Fig. 3)."

First, tiny typo, focuSed and not focuSSed.

Same comment as for Ksg, I believe that the panel 2a doesn't provide much insight and could/should be removed from the main figures.

The authors write "Interestingly, the binding position observed here on the E. coli 30S is distinct from that observed on the T. thermophilus 30S"

This is a very interesting observation! although it is not incumbent to the authors to validate the interpretation of the Ede-70S from T. thermophilus, it would be extremely useful to go back to the experimental data of the latter structure and attempt through a figure (or panel) to validate/verify the authors' interpretation, as the very partial overlapping in terms of binding of Ede between these two different 70S is simply overwhelming and, in theory at least I see no reason why in such different conformations the Ede would bind at the same site!

In "Conservation of the Ede binding site on the 30S subunit"

The authors write "The high resolution of the cryo-EM map of the Ede-30S complex enables a precise description of both direct and indirect water-mediated interactions (Fig. 2e-f and Video S2)"

Same comment as for Ksg, attempt to condense both panels in one.

The authors write "The overall binding mode of Ede is also consistent with the protection of G693 and C795 from chemical modification in the presence of Ede."

I understand the meaning but rephrase and remove the redundancy.

In "Ede overlaps the mRNA path and indirectly inhibits initiator tRNA binding"

The authors write "Moreover, we observe no overlap between the binding position of Ede and that of the accommodated initiator tRNA at the P-site of the 30S (Fig. 2i), leading us to conclude that the observed inhibition by Ede of P-tRNA binding operates indirectly through perturbation of the path of the mRNA, rather than by directly and sterically blocking P-tRNA binding, as proposed previously"

This observation that the initiator tRNA isn't compatible with the binding of Ede is to some extent contradictory with the observation of the authors' derived from 3D particle sorting. Indeed, the authors find a class of particles displaying the presence of an initiator tRNA. If the binding of the latter isn't compatible with that of Ede, how could such a class exist in the dataset? do the authors see Ede in this tRNA-bound partial IC?

In "Cryo-EM structures of GE-30S initiation complexes"

The authors write "Further sorting of the subpopulation with initiator tRNA revealed two distinct classes, with and without the additional presence of IF2 (Supplementary Fig. 5), which were refined to average resolutions of 2.9 Å and 2.8 Å, respectively."

It would be useful to display the final 3D reconstructions including IF2 at a lower threshold in order to fully appreciate the binding of the initiator tRNA and/or IF2, as the intensity of these partners will indicate their level of flexibility.

The authors write "As seen for the Ksg- and Ede-30S-IF1-IF3 structures without initiator tRNA (Fig. 1a, 2a), the 30S-body was well-defined for the GE-30S structure, whereas the head appeared highly flexible (Fig. 3a and Supplementary Fig. 6)."

Same as for Ede and Ksg, panel 3a doesn't provide much more insight compared to panel 3B, and could be, therefore, removed from the main figures.

The authors write “The density was consistent with the revision of the stereochemistry for 3-hydroxy-pipecolic acid moiety of GE30 (Fig. 3c), where one carbon atom had an inverse configuration compared to that reported originally”
It would be best to use the term “puckering”, as it's the latter that is changed and not ONLY one carbon atom.

In “Interaction of GE with the E-site of the 30S subunit”

The authors write “We observe that GE binds in the E-site where it makes extensive direct and water-mediated interactions with nucleotides of the 16S rRNA (Fig. 3e,f and Video S3).”
Same as for Ede and Ksg, try to condense both, the direct and water mediated interactions, in one panel.

In “Ksg, Ede and GE inhibit 30S-IC formation”

The authors write “For Ksg and Ede, the density for the head of the 30S subunit as well as the initiator tRNA were poorly ordered, indicating high flexibility (Supplementary Fig. 2, 4), which precluded models being generated.”
It is certainly unnecessary to model the 30S head, as numerous X-ray and cryo-EM structures of this part of the small subunit do exist for numerous organisms and it's unlikely to find anything new in these structures regarding this part of the 30S. However, it's always possible to locally refine the 30S, but more importantly, thanks to the focused refinement that implicitly includes an analysis of movements en oeuvre, it's possible to tell something about the direction of movements of the 30S head in either of the presented structures, which could be interesting. Moreover, it is not uninteresting to show the exact state of the mRNA channel with regards to its opening or closing, operated by the head.

Reviewer #2

(Remarks to the Author)

Safdari et al. have reported cryo-EM structures of three antibiotics, kasugamycin, edeine, or GE81112, with 30S ribosome initiation intermediate complexes at 2.0 – 2.9 Å resolution. This study provides the molecular mechanism for translation inhibition by these antibiotics.

Kasugamycin is a well-studied drug, and its structures with 30S ribosomes are available. The authors have solved the structure of Ksg in a complex of 30S ribosome, IF1, IF3, and mRNA. However, Ksg also binds to the same binding sites without initiation factors, as reported earlier.

Edeine binds to a similar binding pocket reported earlier in yeast 80S ribosomes. But at a different position compared to the *T. thermophilus* 30S ribosome, as reported earlier.

As reported earlier in the *T. thermophilus* 30S ribosome, GE81112 binds to a distinct site in the *E. coli* 30S ribosome.

Overall, the cryo-EM structures illustrated the detailed interactions of antibiotics, including water-mediated interactions, in the binding pocket of the 30S subunit, which will be crucial for the further design of antibiotic analogs to improve efficacy. The authors have proposed a general mechanism by which these antibiotics alter the mRNA binding, which disfavors translation initiation. The manuscript is nicely written. However, this study incrementally advances the knowledge about the action of these antibiotics.

It would be interesting to compare the structures of antibiotics with and without tRNA and see if the mRNA path and the antibiotic interactions are the same or altered in the presence of the tRNA-bound structure.

Minor comments

1. Supplementary Fig. 1e and Supplementary Fig. 3e. I think, mRNA is missing in the label.
2. In methods ‘30S subunits were re-activated with 21 mM MgCl₂’, an elaboration of why re-activated would be quite useful to the readers.
3. The last lines of the discussion —a juxtaposition of Ksg and GE — hybrid compound— may not be appropriate.

Reviewer #3

(Remarks to the Author)

The work by Safdari et al presents cryo-EM structures of 30S initiation complexes (ICs) in the presence of three translation inhibitors kasugamycin (Ksg), edeine (ED) and GE81112 (GE) respectively. In the major population of each drug-bound 30S IC without the presence of initiator tRNA, the authors resolved the interactions between the drugs with 16S rRNA and mRNA. While the Ksg binding site is consistent with previously reported, ED and GE binding sites differ from previously reported binding sites on the bacterial ribosome, but more consistent with the eukaryotic ribosome. All three drugs are located at the E-site of 30S IC. Based on the impact of the drugs on the path of the bound mRNAs, the authors proposed that the drugs mainly impact the binding of P-site initiator tRNAs through distortion of the mRNAs. Within the minor population of each 30S IC complex, the authors found that for the cases of Ksg and ED, the 30S head is in the open conformation, and the initiator tRNA is in an unaccommodated state; whereas for the case of GE, the tRNA is accommodated, but the required conformational changes for IF3 for its dissociation are not observed. Based on this, the authors proposed that Ksg and ED

further impact initiator tRNA accommodation, but GE impair IF3 dissociation.

Overall the cryo-EM structures are clearly presented. However, the kinetic data for subunit joining and bulk FRET assay lack clear explanation and interpretation to provide additional mechanistic support.

1. The authors interpreted the discrepancy with previous structures for ED and GE as the drug being mis-modeled in the previous structures due to lower resolution or crystal packing artifact. I feel stronger evidence is needed to prove previous results are wrong.
2. The light scattering data for subunit joining measurement does not add much to the structure data interpretation. The lack of detailed analysis on the kinetic parameters prevents the readers from understanding how different inhibition mechanisms by these drugs are can be learnt from the kinetic data.
3. The bulk FRET assay is not described at all in the main text. In the method, the authors only explain the donor and acceptor dyes were placed on different domains of IF3. Presumably the FRET signal is used to monitor the conformational changes of IF3 in the presence of the drug. 30S-IFDL is not explained. In addition, ED and GE both show a smaller amplitude compared to the WT IF3 case (without drug). Therefore, it is unclear how this data is supporting that GE specifically impacts IF3 conformational changes, but not ED.

Reviewer #4

(Remarks to the Author)

The manuscript by the groups of Milon and Wilson entitled "The translation inhibitors kasugamycin, edeine and GE81112 target distinct steps during 30S initiation complex formation" reports high resolution cryo-EM structures (2.0-2.9Å) of 30S initiation complexes bound to three translation initiation inhibitors, kasugamycin (Ksg), edeine (Ede), or GE81112 (GE). These inhibitors have been long known to interfere with the formation of elongation-competent ribosomes. While the binding site of Ksg agrees with previous studies, the current manuscript clarifies the binding modes of Ede and GE, both of which drastically differ from previous reports. The three inhibitors bind to the E site of the 30S subunit and affect distinct steps of 30S initiation complex formation. Ksg disturbs the path of mRNA and indirectly affects the binding of initiator tRNA in the P site. In the 30S-Ede complex, the absence of density for the mRNA in the A, P, and E sites provides a basis for the mode of translation initiation inhibition by Ede. Similar to Ede, GE also affects mRNA binding into the A, P, and E sites, thereby indirectly interfering with accommodation of initiator tRNA into the P site. GE allows the head domain of the 30S subunit to adopt the closed conformation with the accommodated initiator tRNA base paired to the AUG codon in the P site. Still, GE interferes with the path of mRNA preventing initiator tRNA from displacing the C-terminal domain of IF3, blocking joining of the 50S subunit. The structural data is corroborated by rapid kinetics assays measuring the formation of 70S initiation complexes from 30S subunits, initiation factors, initiator tRNA, mRNA, and 50S subunits in the absence or presence of antibiotics, showing that GE interfered to a lesser extent with 70S-IC formation than Ksg and Ede. The work is well executed and is an important step forward in the characterization of translation initiation inhibitors. The high quality structures presented will guide the development of improved therapeutics.

From the viewpoint of this reviewer, the manuscript could be accepted in its current form. Yet, addressing the following points may improve the presentation.

Although the structure of GE81112 bound to the E. coli 30S initiation complex reported in 2017 by Lopez-Alonso et al. (NAR) had a low resolution (13.5 Å), they had reasonably outlined from their 'locked' and 'unlocked' 30S-PIC complexes that the mechanism of inhibition was due to the unaccommodated tRNA in the P site. This must be referenced properly (Reference 28 is wrong). Further, there is a new bioRxiv preprint (Sept. 26, 2024) reporting 30S-PIC structures with GE81112, which could be referenced (<https://www.biorxiv.org/content/10.1101/2024.09.26.614503v1.full>).

On page 12, the authors write "No differences were observed in the GE-30S complexes with or without IF2" What does this mean? We can presume that initiator tRNA in the IF2-bound complex adopts the same conformation as that seen in the GE complex without IF2. In the IF2-containing complex, does domain C2 of IF2 interact with the fMet residue of initiator tRNA? One would expect the p/l conformation of initiator tRNA in the presence of IF2. Could GE be interfering with the p/l conformation of initiator tRNA? Additional discussion and possibly showing a model for the IF2-containing 30S-IC-GE complex (Suppl. Fig. 5i) would be beneficial (if density allows to build a model).

For completeness, it may be useful to mention that while the overall inhibition by Ksg occurs by disturbing the path of mRNA and indirectly affecting the binding of initiator tRNA in the P site, a subset of genes also continues to be translated. A recent ribosome profiling study with Ksg-treated E. coli showed translation of full-length proteins in the presence of saturating concentrations of Ksg (Zhang and Mankin et al., PNAS 2022). The authors could discuss this aspect mentioning that it will require additional structural studies to elucidate the mechanism of codon/mRNA sequence specificity of Ksg.

Minor edits:

S.Fig.1 – legend f, 'The latter two classed' change to 'classes'
Spelling errors 'modelled', 'focussed' to be corrected

In Fig. 2e, f, the guanylspermidine tail of Ede overlaps with the rest of the Ede model. May use a different color/view? Furthermore, it is unclear in Fig. 2e which nucleotide is G926.

Page 3, last sentence, "However, to date no structures of Ksg bound..." would be better as "However, to date there are no structures of Ksg..."

Page 4, "...GE interacts with and distorts the anticodon stem-loop of an initiator tRNA." replace "an" with "the" or remove article altogether.

Page 4, "...but the resolution 13.5 Å..." to "...but the 13.5 Å-resolution..."

Page 6, "Shine-Dalgarno" is Shine-Dalgarno

Page 6, "...and accommodated initiator-tRNA..." delete hyphen.

Page 6, "consistent with the proposal that Ksg interferes with initiator fMet-tRNA^{fMet} tRNA binding indirectly by perturbing the placement of the mRNA" should be "consistent with the proposal that Ksg indirectly interferes with initiator tRNA binding..."

Page 8, "the binding site may crystal packing artefacts..." should be "the binding site may be distorted due to crystal packing artefacts..."

Page 11, "the -1 and -2 nucleotides of the E-site codon of the mRNA (Fig. 3g-i)." Mentioning from where the mRNA is taken from would make it clearer, e.g. from a previous 30S-IC complex (ref. 6).

Page 13, "stabilize the binding of IF3 on the 30S and by reducing its exchange rate (Fig. 4h)." Delete the word "and".

Page 14, "On one hand,..." should be "On the one hand, ..."

Page 14, "lends itself to the generation of hybrid compounds formed by linking these two compounds." Repeat of word "compounds" replace one with "drugs"?

Page 25, "Preparation of cryo-EM grids and data collection" change to "Preparation of cryo-EM grids"

Version 1:

Reviewer comments:

Reviewer #1

(Remarks to the Author)

The authors have clarified appropriately the raised questions and addressed any remaining concerns.

Reviewer #2

(Remarks to the Author)

The authors have addressed my queries adequately. They have elaborated on 30S subunit re-activation with 21 mM MgCl₂ before forming its complexes with initiation factors and antibiotics.

Reviewer #3

(Remarks to the Author)

The authors have improved the explanation of the light scattering and FRET part of the manuscript. However, there are additional minor issues that the authors should address:

(1) Please show the fitting results of the light scattering data and the FRET data in a supplemental table. The light scattering data is fit with a biphasic exponential model. When comparing the scattering data with and without antibiotics, the efficiency and the apparent rate are used. Are the apparent rates reported as the average of rates from the two-phasic association? The fold change is not obvious from the traces alone.

(2) The FRET data is still confusing. Particularly IF3DL is referred to as the double-labeled IF3, which presumably only contains mutations necessary for the labeling, and serves as the "WT" IF3 in the FRET measurement, which should be equivalent to IF3 wt in Fig 4h. Then I don't quite understand what 30S-IF3DL is representing. The authors should clarify the labels in the figures in the figure caption to increase the readability.

(3) In the FRET measurement, the authors claim that EDE and GE81112 affect IF3 dissociation, but not KSG. However, in the KSG curve, the IF3 dissociation appears to be more efficient. Is the different with the experimental error? The standard deviation from replicates should be reported in the SI table including all the fitting results for quantitative comparison.

Reviewer #4

(Remarks to the Author)

The authors have appropriately addressed my concerns and the manuscript is now ready for publication.

Reviewer #1 (Remarks to the Author):

The manuscript by Haaris et al., from the Wilson group, reports the results of a structural and functional investigation aiming at understanding the action mode of three known antibiotics that act on the 30S (small ribosomal subunit of the bacterial ribosome), Kasugamycin, Edeine and GE81112. The authors mainly present structures of the initiation complexes (including the full set of initiation factors reconstituted *in vitro*) bound to these antibiotics, one at a time. The structures reveal that all three antibiotics bind within the E-site of the 30S and disadvantage 30S initiation complex formation. It has been previously demonstrated that kasugamycin (Ksg) and edeine (Ede) affect early steps of 30S pre-initiation complex formation, while GE81112 (GE) stalls pre-initiation complex formation at a later step by impeding IF3 departure after the start codon recognition. However, until the current study, no structures of Ksg, Ede and/or GE bound to translation initiation complexes have been determined.

The determined cryo-EM structures of Ksg, Ede and GE within the context of *E. coli* 30S (intermediate) initiation complexes were solved at resolutions varying between 2.0 and 2.9 Å. The authors show that all three antibiotics inhibit translation initiation by interfering with 30S-IC assembly and therefore precluding 70S-IC formation and mRNA translation.

The manuscript is well written and mostly pretty clear, with few exceptions. The figures are all required for the proper understanding of the manuscript and are, in addition, aesthetic. However, some effort could be made to streamline the figure a bit further (see reviewer's comments below), which remains totally optional and to the discretion of the authors.

The reviewer only has few minor points:

Minor points:

In Results:

In "Cryo-EM structures of Ksg-30S initiation complexes"

The authors write "In silico sorting revealed one major population (359,652 particles, 76%) comprising 30S subunits with mRNA, IF1 and IF3, as well as a second minor population (61,790 particles, 13%) with the additional presence of initiator tRNA (Supplementary Fig. 1). These two subpopulations, with and without initiator tRNA, could be refined to average resolutions of 2.9 Å and 2.5 Å, respectively (Supplementary Fig. 2)."

Figure S2 shows that the tRNA-dissociated class refined down to 2.4 and not 2.5 Å.

We would like to clarify that 2.5 Å is the resolution that we obtained before focused refinement which is mentioned in the text that reviewer has highlighted and also depicted in in silico sorting scheme of E. coli Kgs-30S complex depicted in Sup Fig 1. We have stated further that "we performed focused refinement on the 30S-body, which further improved the resolution (2.4 Å) and density quality for the 30S-body, IF1 and IF3 (Fig. 1b and Supplementary Fig. 2)".

The authors write "Therefore, we performed focussed refinement on the 30S-body,

which further improved the resolution (2.4 Å) and density quality for the 30S-body, IF1 and IF3 (Fig. 1b and Supplementary Fig. 2).“

The interest of showing the non-locally refined structure in panel 1a is very limited, I suggest its removal from the main figures.

We would still like to retain the non-locally refined structure due to two reasons. Firstly, the low resolution of the head in this reconstruction shows the dynamics of head and secondly, at this lower resolution the density for the mRNA is seen, which becomes weaker or lost in the post-processed maps shown in panel b. Moreover, it might be little confusing for the readers to directly see a reconstruction without a head. In contrast, our present scheme highlights that head has been removed (Fig. 1b) from the map.

The authors write “The quality of the cryo-EM map enabled a precise description of direct and indirect water-mediated interactions between Ksg and the 16S rRNA (Fig. 1e,f and Video S1).“

I believe it's straightforward to show both kinds of interactions on the same panel, direct and indirect water mediated. therefore, I suggest the removal of one of the panels, the authors could keep panel f, for example, and add the direct interactions on the top.

We initially thought about depicting both direct and indirect interaction in same panel and tried to do this, but it quickly became apparent that it is difficult to clearly distinguish the direct and indirect interaction when present in the same image since in many cases they are overlapping with each other. Therefore, to avoid this confusion, and maintain clarity, we would prefer to keep the direct and indirect interactions as separate panels.

In “Ksg alters the path of the mRNA through the A-, P- and E-sites”

The authors write “Although the additional density was partially fragmented, strong regions of density were observed adjacent to 16S rRNA nucleotides G693, A790, G926 and C1400, suggesting stacking interactions of mRNA nucleotides with these bases (Fig. 1g, ED Fig. 1e).“

The low resolution of the non-locally refined map makes difficult, and to some extent risky, to interpret these bits of mRNA densities, especially since their structure and precise orientation don't provide much of relevant insight beyond the presence of mRNA in its channel.

We thank the reviewer for highlighting this important point. Indeed, the bits of mRNA densities are not at a resolution where we can confidently assign the identity of the bases. However, we would like to reiterate that we only tentatively modelled the bases to depict the possible numbering of those bases in relation to a comparison with structures previously deposited (PDB:5LMN). We were definitely able to confidently model nucleotides G693, A790, G926 and C1400 and we just state that we were able to see densities of mRNA which stack on these bases.

The authors write “Therefore, we generated a tentative model for the path of the mRNA in the presence of Ksg (Fig. 1g), which was similar, but distinct, from the path reported in a previous structure of 30S-IC lacking initiator tRNA (Fig. 1h and ED Fig. 1f)” Panel 1h is confusing as a main figure, as it could be seen as conveying the message that Ksg is incompatible with the binding of mRNA, which is the opposite of the message intended by the authors.

We think it is important to keep panel 1h since we use this mRNA to propose the register for the mRNA that we observe in the presence of Ksg. Although the clash may cause some confusion, we think it is honest to show this clearly. It could be that different mRNAs adopt different conformations on the 30S, but it could also be due simply to the relatively low resolution of the previous reconstructions. For this reason, we do not make a point about this potential steric clash.

In “Cryo-EM structures of Ede-30S initiation complexes”

The authors write “As seen for the Ksg-30S structure without initiator tRNA (Fig. 1a), the 30S-body was well-defined for the Ede-30S structure without initiator tRNA, whereas the head was poorly resolved (Fig. 2a and Supplementary Fig. 4). Focussed refinement further improved the resolution (to 2.0 Å) and quality of the cryo-EM map density for the 30S-body, IF1 and IF3 (Fig. 2b and Supplementary Fig. 3).”

First, tiny typo, focuSed and not focuSSed.

Focussed is not a typo. It is the Oxford English spelling for the word focussed, rather than the American spelling focused.

Same comment as for Ksg, I believe that the panel 2a doesn't provide much insight and could/should be removed from the main figures.

As for Ksg, we would like to retain both panel 2a and panel 2b due to reasons discussed above.

The authors write “Interestingly, the binding position observed here on the *E. coli* 30S is distinct from that observed on the *T. thermophilus* 30S” This is a very interesting observation! although it is not incumbent to the authors to validate the interpretation of the Ede-70S from *T. thermophilus*, it would be extremely useful to go back to the experimental data of the latter structure and attempt through a figure (or panel) to validate/verify the authors' interpretation, as the very partial overlapping in terms of binding of Ede between these two different 70S is simply overwhelming and, in theory at least I see no reason why in such different conformations the Ede would bind at the same site!

*We assume the reviewer refers to the structure of edeine on the *T. thermophilus* 30S subunit and not on a 70S ribosome as stated. In fact, we have dedicated panels ED4a-d to depict how the model of edeine in *T. thermophilus* 30S ribosome clashes with their own deposited model and also other ribosome structures on alignment.*

Therefore, this binding position cannot exist and we do not think it makes sense to go back to the original data to explore this further. By contrast, the binding site of edeine in our structure is very similar to the one determined on yeast 80S ribosome.

In “Conservation of the Ede binding site on the 30S subunit”

The authors write “The high resolution of the cryo-EM map of the Ede-30S complex enables a precise description of both direct and indirect water-mediated interactions (Fig. 2e-f and Video S2)”

Same comment as for Ksg, attempt to condense both panels in one.

As for Ksg, we would like to retain both panel 2a and panel 2b due to reasons discussed above.

The authors write “The overall binding mode of Ede is also consistent with the protection of G693 and C795 from chemical modification in the presence of Ede.” I understand the meaning but rephrase and remove the redundancy.

We thank the reviewer for the comment. The text now reads “The binding mode of Ede is consistent with chemical probing assays where edeine protects G693 and C795 from chemical modification”.

In “Ede overlaps the mRNA path and indirectly inhibits initiator tRNA binding”

The authors write “Moreover, we observe no overlap between the binding position of Ede and that of the accommodated initiator tRNA at the P-site of the 30S (Fig. 2i), leading us to conclude that the observed inhibition by Ede of P-tRNA binding operates indirectly through perturbation of the path of the mRNA, rather than by directly and sterically blocking P-tRNA binding, as proposed previously”

This observation that the initiator tRNA isn't compatible with the binding of Ede is to some extent contradictory with the observation of the authors' derived from 3D particle sorting. Indeed, the authors find a class of particles displaying the presence of an initiator tRNA. If the binding of the latter isn't compatible with that of Ede, how could such a class exist in the dataset? do the authors see Ede in this tRNA-bound partial IC?

*We have shown that the majority of particles in the Ede dataset (80.2%) do not have tRNA. Also, in these complexes, Ede perturbs with the path of mRNA leading to prevention of tRNA binding, rather than directly sterically blocking tRNA as reported earlier. However, this is not 100% efficient and indeed we do find some particles (12.6%) containing tRNA. The point is that Ede does not allow **full** accommodation of the tRNA and thus in the presence of Ede, the tRNA is not in conformation that allows base pairing with anticodon of tRNA. This is already mentioned in the text:*

The observation that the major subpopulations of the Ksg-, Ede- and GE-30S complexes lacked tRNA is consistent with the suggestion that the drugs interfere with initiator tRNA binding and/or accommodation in the P-site indirectly.

Nevertheless, for all three antibiotics, we observed minor subpopulations (7-13%) of 30S complexes that contained initiator tRNA.

In “Cryo-EM structures of GE-30S initiation complexes”

The authors write “Further sorting of the subpopulation with initiator tRNA revealed two distinct classes, with and without the additional presence of IF2 (Supplementary Fig. 5), which were refined to average resolutions of 2.9 Å and 2.8 Å, respectively.” It would be useful to display the final 3D reconstructions including IF2 at a lower threshold in order to fully appreciate the binding of the initiator tRNA and/or IF2, as the intensity of these partners will indicate their level of flexibility.

We now display the final 3D reconstructions including IF2 at a lower threshold in Supplementary Fig 5j (now Supplementary Fig 9j).

The authors write “As seen for the Ksg- and Ede-30S-IF1-IF3 structures without initiator tRNA (Fig. 1a, 2a), the 30S-body was well-defined for the GE-30S structure, whereas the head appeared highly flexible (Fig. 3a and Supplementary Fig. 6).” Same as for Ede and Ksg, panel 3a doesn't provide much more insight compared to panel 3B, and could be, therefore, removed from the main figures.

As for Ksg and Ede, we would like to retain both panel 3a and panel 3b due to reasons discussed above.

The authors write “The density was consistent with the revision of the stereochemistry for 3-hydroxypipicolinic acid moiety of GE30 (Fig. 3c), where one carbon atom had an inverse configuration compared to that reported originally” It would be best to use the term "puckering", as it's the latter that is changed and not ONLY one carbon atom.

The reviewer is correct that the inverse configuration leads to a change in the puckering. We have now incorporated this point into the text of the revised version

In “Interaction of GE with the E-site of the 30S subunit”

The authors write “We observe that GE binds in the E-site where it makes extensive direct and water-mediated interactions with nucleotides of the 16S rRNA (Fig. 3e,f and Video S3).”

Same as for Ede and Ksg, try to condense both, the direct and water mediated interactions, in one panel.

As discussed earlier for Ksg, we would like to keep direct and water-mediated interactions separate to maintain clarity.

In “Ksg, Ede and GE inhibit 30S-IC formation”

The authors write “For Ksg and Ede, the density for the head of the 30S subunit as

well as the initiator tRNA were poorly ordered, indicating high flexibility (Supplementary Fig. 2, 4), which precluded models being generated.“ It is certainly unnecessary to model the 30S head, as numerous X-ray and cryo-EM structures of this part of the small subunit do exist for numerous organisms and it's unlikely to find anything new in these structures regarding this part of the 30S. However, it's always possible to locally refine the 30S, but more importantly, thanks to the focused refinement that implicitly includes an analysis of movements en oeuvre, it's possible to tell something about the direction of movements of the 30S head in either of the presented structures, which could be interesting. Moreover, it is not uninteresting to show the exact state of the mRNA channel with regards to its opening or closing, operated by the head.

We thank the reviewer for this comment. Indeed, the proposed analysis could unveil unknown networks of head to body communication. A rapid estimation of the particle number required to efficiently (in terms of time and computational power) obtain relevant data exceeds the current datasets. We are at moment setting up such analysis for other purposes, if successful, it would be appealing to revisit the mechanisms described here from a more dynamic perspective.

Reviewer #2 (Remarks to the Author):

Safdari et al. have reported cryo-EM structures of three antibiotics, kasugamycin, edeine, or GE81112, with 30S ribosome initiation intermediate complexes at 2.0 – 2.9 Å resolution. This study provides the molecular mechanism for translation inhibition by these antibiotics.

Kasugamycin is a well-studied drug, and its structures with 30S ribosomes are available. The authors have solved the structure of Ksg in a complex of 30S ribosome, IF1, IF3, and mRNA. However, Ksg also binds to the same binding sites without initiation factors, as reported earlier.

Edeine binds to a similar binding pocket reported earlier in yeast 80S ribosomes. But at a different position compared to the T. thermophilus 30S ribosome, as reported earlier.

As reported earlier in the T. thermophilus 30S ribosome, GE81112 binds to a distinct site in the E. coli 30S ribosome.

Overall, the cryo-EM structures illustrated the detailed interactions of antibiotics, including water-mediated interactions, in the binding pocket of the 30S subunit, which will be crucial for the further design of antibiotic analogs to improve efficacy. The authors have proposed a general mechanism by which these antibiotics alter the mRNA binding, which disfavors translation initiation. The manuscript is nicely written. However, this study incrementally advances the knowledge about the action of these antibiotics.

We respectfully disagree that the study “incrementally advances the knowledge about the action of these antibiotics”. Our study identifies completely different binding sites for edeine and GE81112 on the 30S subunit, compared to previous studies on bacterial ribosomes. Moreover, all previous structures were at lower resolution

and/or were not in the context of physiological initiation complexes. Therefore, we feel it makes a very important advance in our understanding of these antibiotics.

It would be interesting to compare the structures of antibiotics with and without tRNA and see if the mRNA path and the antibiotic interactions are the same or altered in the presence of the tRNA-bound structure.

For Ksg and Ede tRNA states, the mRNA is too poorly resolved to make such a comparison. For the GE tRNA state, we have modelled and show the mRNA density in the movie. However, in the case of GE, we do not observe density for the mRNA in the absence of the tRNA, therefore, such a comparison of the mRNA path in the presence and absence of tRNA is not possible.

Minor comments

1. Supplementary Fig. 1e and Supplementary Fig. 3e. I think, mRNA is missing in the label.

We have corrected this.

2. In methods '30S subunits were re-activated with 21 mM MgCl₂', an elaboration of why re-activated would be quite useful to the readers.

30S subunit reactivation is a well-established step in ribosome biochemistry, dating back to the foundational work of Pestka and Nirenberg in the 1960s (JMB 1966, PMID: 136111). Around the same time, Nomura demonstrated that raising the magnesium concentration restored the translational activity of 30S subunits reconstituted from their component parts. Lowering magnesium levels during purification may destabilize key ionic interactions, particularly in the decoding center and helix 44 (h44), leading to diminished translational competence.

Contributions from the Noller group indicated that universally conserved nucleotides critical for decoding are directly involved in this reactivation process (JMB 1986, PMID: 2434656). More recently, the Ortega group provided a cryo-EM structure of the inactive 30S subunit, showing that reduced magnesium conditions can induce alternative base-pairing in h44 and at the 3' end of the 16S rRNA (RNA 2020, PMID: 32989043). Notably, these conformational changes are reversible: restoring the magnesium concentration allows the subunit to regain its functional structure.

Taken together, these findings underscore that increasing magnesium concentration following purification is necessary to return 30S subunits to an active state. This step promotes that essential ions are restored to their proper positions, stabilizing the decoding center and h44 and thereby restoring full translational functionality.

We added the following sentence to the methods section: "Raising the magnesium concentration (21 mM MgCl₂) restores essential ionic interactions and re-activates purified 30S subunits, as previously demonstrated by classical and more recent structural studies in ribosome biochemistry."

3. The last lines of the discussion—a juxtaposition of Ksg and GE—hybrid compound—may not be appropriate.

We have rephrased the text here now to read: “In this regard, we note the binding positions of Ksg and GE (Fig. 5a) lend themselves to the generation of hybrid compounds formed by linking these two drugs.”

Reviewer #3 (Remarks to the Author):

The work by Safdari et al presents cryo-EM structures of 30S initiation complexes (ICs) in the presence of three translation inhibitors kasugamycin (Ksg), edeine (ED) and GE81112 (GE) respectively. In the major population of each drug-bound 30S IC without the presence of initiator tRNA, the authors resolved the interactions between the drugs with 16S rRNA and mRNA. While the Ksg binding site is consistent with previously reported, ED and GE binding sites differ from previously reported binding sites on the bacterial ribosome, but more consistent with the eukaryotic ribosome. All three drugs are located at the E-site of 30S IC. Based on the impact of the drugs on the path of the bound mRNAs, the authors proposed that the drugs mainly impact the binding of P-site initiator tRNAs through distortion of the mRNAs. Within the minor population of each 30S IC complex, the authors found that for the cases of Ksg and ED, the 30S head is in the open conformation, and the initiator tRNA is in an unaccommodated state; whereas for the case of GE, the tRNA is accommodated, but the required conformational changes for IF3 for its dissociation are not observed. Based on this, the authors proposed that Ksg and ED further impact initiator tRNA accommodation, but GE impair IF3 dissociation.

Overall the cryo-EM structures are clearly presented. However, the kinetic data for subunit joining and bulk FRET assay lack clear explanation and interpretation to provide additional mechanistic support.

We appreciate your feedback and apologize for the earlier oversight. We have now clarified the connection between the structural and kinetic data in the revised text. Specifically, we highlight how the subunit-joining (SJ) and bulk FRET assays, when viewed alongside our detailed cryo-EM reconstructions, provide a cohesive narrative that supports our proposed model. The structural snapshots alone do not suffice to establish a precise mechanism for how each compound acts. However, by incorporating the temporal resolution offered by the SJ and FRET assays, we can more confidently infer the distinct stages where these compounds exert their influence. Taken together, these complementary approaches strengthen our conclusions and provide a more robust mechanistic framework.

1. The authors interpreted the discrepancy with previous structures for ED and GE as the drug being mis-modeled in the previous structures due to lower resolution or crystal packing artifact. I feel stronger evidence is needed to prove previous results are wrong.

*We agree that ultimately structures of Ede and GE on *T. thermophilus* ribosomes will be necessary to address this more comprehensively, but we feel that this is out of*

the scope of our study. However, we do feel that the current evidence is already sufficiently strong to cast quite some doubt on the previous studies:

Our cryo-EM structures of edeine and GE bound to the E. coli 30S subunit are resolved at 2.0–2.9 Å, which is significantly better than the resolution of previously reported structures, such as the 4.5 Å edeine structure on T. thermophilus 30S and the 3.5 Å GE structure on T. thermophilus 30S. This resolution allowed us to unambiguously assign the binding sites of both antibiotics in the E-site, a location distinct from that previously reported. The precise placement of drugs, along with their detailed interactions with rRNA nucleotides and water-mediated interaction, is visible in our maps, leaving little room for alternative interpretations.

Additionally, our findings align with the earlier reported biochemistry. For example, the overall binding mode of Ede is consistent with the protection of G693 and C795 from chemical modification by the presence of Ede while previous structure of Ede on T. thermophilus is at odds with this data. Similarly, the interaction of the 5AH moiety of GE with G693 in our structure is likely to explain the protection of this base from modification by GE as reported earlier. By contrast, the previous binding site of GE cannot explain this protection.

Our Ede structure on E.coli ribosome highlights the conserved binding mode of edeine in the E-site across bacterial and eukaryotic ribosomes. This conservation strongly supports the physiological relevance of the binding site we observed, contrasting with the unique and inconsistent positioning reported previously for Ede on T. thermophilus ribosome.

Finally, we would like to highlight a recent work which came out as preprint nearly same time as our manuscript reporting 30S-PIC cryo-EM structures with GE81112, which also reports the same binding site of GE as our structures (<https://www.biorxiv.org/content/10.1101/2024.09.26.614503v1.full>).

Nevertheless, we have now changed the text related to Ede to state that this evidence “leads us to suggest” rather than “suggesting...”. For GE, we had anyway stated that “it is possible that GE81112B has a distinct binding site on T. thermophilus 30S as reported previously{Fabbretti, 2016 #18054}” but that “we believe that this may have arisen due to crystal packing artefacts within the T. thermophilus 30S crystals.”

2. The light scattering data for subunit joining measurement does not add much to the structure data interpretation. The lack of detailed analysis on the kinetic parameters prevents the readers from understanding how different inhibition mechanisms by these drugs are can be learnt from the kinetic data.

We apologize for this lack of clarity and have now revised the text to read: “To directly investigate whether GE, as well as Ksg and Ede, compromises 70S-IC formation, we assembled 30S initiation complexes in the absence and presence of each antibiotic and, upon addition of 50S, 70S-IC formation was monitored using light scattering in a stopped-flow instrument^{1,33} (Fig. 4g). In the absence of the drugs, addition of 50S led to a rapid increase in light scattering over time, indicating 70S-IC formation. The time traces were biphasic consistent with a two-step mechanism of 50S

binding to 30S ICs^{1,9}. The presence of Ksg and Ede interfered with 50S joining (Fig. 4g) by reducing (over 5-fold), the efficiency (as seen from the amplitude) and apparent rates of 50S binding, consistent with their ability to lock translation initiation in an early 30S-PIC state, likely, with the open head and unaccommodated initiator tRNA (Fig. 4a,b). Similarly, GE also decreased 50S joining kinetics by over 5-fold, however, the 70S formation efficiency, as judged by the amplitude of the scattered light, was affected only by a factor of two (Fig. 4g). GE presumably allows the 30S to adopt a more 30S-IC-like state, consistent with the compound allowing start codon recognition; yet, reducing the kinetics of IF3 displacement as shown in the structures above (Fig. 3). All three compounds share a similar binding site as that of IF3; thus, to study their direct effect on the 30S-bound IF3, we performed experiments monitoring the exchange of IF3 on the 30S subunit. We used a double-labelled derivative of IF3 (IF3DL) to monitor factor dissociation by intramolecular FRET⁸. IF3DL binding to the 30S results in donor fluorescence increase, while the opposite is valid for factor dissociation. We observe that GE and Ede stabilize IF3 on the 30S as observed by a 3-fold reduction of the factor exchange rate. On the other hand, Ksg does not perturb IF3 dissociation rate constant from the ternary 30S-IF3DL-Ksg complex, suggesting that Ksg, differently from GE and Ede, does not directly affect the early 30S-IF3 complex (Fig. 4h)."

3. The bulk FRET assay is not described at all in the main text. In the method, the authors only explain the donor and acceptor dyes were placed on different domains of IF3. Presumably the FRET signal is used to monitor the conformational changes of IF3 in the presence of the drug. 30S-IFDL is not explained. In addition, ED and GE both show a smaller amplitude compared to the WT IF3 case (without drug). Therefore, it is unclear how this data is supporting that GE specifically impacts IF3 conformational changes, but not ED.

Now a more extensive explanation is provided in the text, see above

Reviewer #4 (Remarks to the Author):

The manuscript by the groups of Milon and Wilson entitled "The translation inhibitors kasugamycin, edeine and GE81112 target distinct steps during 30S initiation complex formation" reports high resolution cryo-EM structures (2.0-2.9Å) of 30S initiation complexes bound to three translation initiation inhibitors, kasugamycin (Ksg), edeine (Ede), or GE81112 (GE). These inhibitors have been long known to interfere with the formation of elongation-competent ribosomes. While the binding site of Ksg agrees with previous studies, the current manuscript clarifies the binding modes of Ede and GE, both of which drastically differ from previous reports. The three inhibitors bind to the E site of the 30S subunit and affect distinct steps of 30S initiation complex formation. Ksg disturbs the path of mRNA and indirectly affects the binding of initiator tRNA in the P site. In the 30S-Ede complex, the absence of density for the mRNA in the A, P, and E sites provides a basis for the mode of translation initiation inhibition by Ede. Similar to Ede, GE also affects mRNA binding into the A, P, and E sites, thereby indirectly interfering with accommodation of initiator tRNA into the P site. GE allows the head domain of the 30S subunit to adopt

the closed conformation with the accommodated initiator tRNA base paired to the AUG codon in the P site. Still, GE interferes with the path of mRNA preventing initiator tRNA from displacing the C-terminal domain of IF3, blocking joining of the 50S subunit. The structural data is corroborated by rapid kinetics assays measuring the formation of 70S initiation complexes from 30S subunits, initiation factors, initiator tRNA, mRNA, and 50S subunits in the absence or presence of antibiotics, showing that GE interfered to a lesser extent with 70S-IC formation than Ksg and Ede. The work is well executed and is an important step forward in the characterization of translation initiation inhibitors. The high quality structures presented will guide the development of improved therapeutics.

From the viewpoint of this reviewer, the manuscript could be accepted in its current form. Yet, addressing the following points may improve the presentation.

Although the structure of GE81112 bound to the E. coli 30S initiation complex reported in 2017 by Lopez-Alonso et al. (NAR) had a low resolution (13.5 Å), they had reasonably outlined from their 'locked' and 'unlocked' 30S-PIC complexes that the mechanism of inhibition was due to the unaccommodated tRNA in the P site. This must be referenced properly (Reference 28 is wrong). Further, there is a new bioRxiv preprint (Sept. 26, 2024) reporting 30S-PIC structures with GE81112, which could be referenced (<https://www.biorxiv.org/content/10.1101/2024.09.26.614503v1.full>).

We thank the reviewer for noticing the incorrect reference 28. This is now corrected to Lopez-Alonso, J.P. et al. Structure of a 30S pre-initiation complex stalled by GE81112 reveals structural parallels in bacterial and eukaryotic protein synthesis initiation pathways. Nucleic Acids Res 45, 2179-2187 (2017). We have also included citation to the new preprint that came out after we had submitted our manuscript that also reports the same binding site of GE on the 30S as reported here and different to that reported previously. This is the new reference 32: Schedlbauer, A. et al. A binding site for the antibiotic GE81112 in the ribosomal mRNA channel. bioRxiv (2024).

On page 12, the authors write "No differences were observed in the GE-30S complexes with or without IF2" What does this mean? We can presume that initiator tRNA in the IF2-bound complex adopts the same conformation as that seen in the GE complex without IF2. In the IF2-containing complex, does domain C2 of IF2 interact with the fMet residue of initiator tRNA? One would expect the p/I conformation of initiator tRNA in the presence of IF2. Could GE be interfering with the p/I conformation of initiator tRNA? Additional discussion and possibly showing a model for the IF2-containing 30S-IC-GE complex (Suppl. Fig. 5i) would be beneficial (if density allows to build a model).

The reviewer is correct that we see that initiator tRNA in the IF2-bound complex adopting the same conformation as that seen in the GE complex without IF2. Unfortunately, we don't have resolution for the C2 domain region of IF2 to ascertain its interaction with initiator tRNA in presence of IF2. Indeed, it is plausible that GE interferes with the p/I conformation of initiator tRNA in the presence of IF2. We think, however, that although the initiator tRNA can base pair with the start codon, the

presence of GE blocks the canonical path of the mRNA in the closed form and prevents the initiator tRNA from displacing the CTD of IF3 from the 30S.

For completeness, it may be useful to mention that while the overall inhibition by Ksg occurs by disturbing the path of mRNA and indirectly affecting the binding of initiator tRNA in the P site, a subset of genes also continues to be translated. A recent ribosome profiling study with Ksg-treated E. coli showed translation of full-length proteins in the presence of saturating concentrations of Ksg (Zhang and Mankin et al., PNAS 2022). The authors could discuss this aspect mentioning that it will require additional structural studies to elucidate the mechanism of codon/mRNA sequence specificity of Ksg.

We thank the reviewer for this comment and have now added a few sentences to the discussion, which reads “Recent studies have shown that Ksg can also interfere with translation after 70S formation, and that the extent of inhibition is modulated by the nature of the mRNA, specifically, the nucleotide directly upstream of the start codon³⁴. It will be interesting in the future to investigate whether similar effects are also observed for Ede and GE, and to provide a structural basis for the context-specificity.”

Minor edits:

S.Fig.1 – legend f, ‘The latter two classed’ change to ‘classes’

Corrected

Spelling errors ‘modelled’, ‘focussed’ to be corrected

No correction is needed here. Modelled and focussed are the Oxford English spelling of these words, rather than the American spelling.

In Fig. 2e, f, the guanylspermidine tail of Ede overlaps with the rest of the Ede model. May use a different color/view? Furthermore, it is unclear in Fig. 2e which nucleotide is G926.

We have slightly rotated the view and shifted the G926 label.

Page 3, last sentence, “However, to date no structures of Ksg bound...” would be better as “However, to date there are no structures of Ksg...”

We have changed the sentence as reviewer suggested.

Page 4, “...GE interacts with and distorts the anticodon stem-loop of an initiator tRNA.” replace “an” with “the” or remove article altogether.

We have removed the article.

Page 4, “...but the resolution 13.5 Å...” to “...but the 13.5 Å-resolution...”

Changed

Page 6, “Shine-Dalgarno” is Shine-Dalgarno

Corrected

Page 6, “..and accommodated initiator-tRNA...” delete hyphen.

Deleted.

Page 6, “consistent with the proposal that Ksg interferes with initiator fMet-tRNA^{fMet} tRNA binding indirectly by perturbing the placement of the mRNA” should be “consistent with the proposal that Ksg indirectly interferes with initiator tRNA binding...”

Corrected.

Page 8, “the binding site may crystal packing artefacts...” should be “the binding site may be distorted due to crystal packing artefacts...”

Corrected.

Page 11, “the -1 and -2 nucleotides of the E-site codon of the mRNA (Fig. 3g-i).” Mentioning from where the mRNA is taken from would make it clearer, e.g. from a previous 30S-IC complex (ref. 6).

Corrected.

Page 13, “stabilize the binding of IF3 on the 30S and by reducing its exchange rate (Fig. 4h).” Delete the word “and”.

This sentence has now been omitted since we have introduced more description of light scattering subunit joining and bulk FRET experiments.

Page 14, “On one hand,...” should be “On the one hand, ...”

Corrected.

Page 14, “lends itself to the generation of hybrid compounds formed by linking these two compounds.” Repeat of word “compounds” replace one with “drugs”?

We have now replaced the latter “compounds” with “drugs” as per the suggestion.

Page 25, “Preparation of cryo-EM grids and data collection” change to “Preparation of cryo-EM grids”

Corrected.

RESPONSE TO REVIEWERS' COMMENTS

Reviewer #1 (Remarks to the Author):

The authors have clarified appropriately the raised questions and addressed any remaining concerns.

Reviewer #2 (Remarks to the Author):

The authors have addressed my queries adequately. They have elaborated on 30S subunit re-activation with 21 mM MgCl₂ before forming its complexes with initiation factors and antibiotics.

Reviewer #3 (Remarks to the Author):

The authors have improved the explanation of the light scattering and FRET part of the manuscript. However, there are additional minor issues that the authors should address:

(1) Please show the fitting results of the light scattering data and the FRET data in a supplemental table. The light scattering data is fit with a biphasic exponential model. When comparing the scattering data with and without antibiotics, the efficiency and the apparent rate are used. Are the apparent rates reported as the average of rates from the two-phasic association? The fold change is not obvious from the traces alone.

Thank you for the annotation. A table has been added as suggested, showing both amplitude and apparent rates (fast and slow) as well as the average rate. In the text, it is now specified that the average rate is used for comparisons.

(2) The FRET data is still confusing. Particularly IF3DL is referred to as the double-labeled IF3, which presumably only contains mutations necessary for the labeling, and serves as the “WT” IF3 in the FRET measurement, which should be equivalent to IF3 wt in Fig 4h. Then I don't quite understand what 30S-IF3DL is representing. The authors should clarify the labels in the figures in the figure caption to increase the readability.

Sorry about the confusion, now the labels are properly indicated. Also the legend now indicates the buffer control for the mixing apparatus (wrongly called 30S-IF3DL), and IF3wt refers to the signal in the absence of the antibiotic.

(3) In the FRET measurement, the authors claim that EDE and GE81112 affect IF3 dissociation, but not KSG. However, in the KSG curve, the IF3 dissociation appears to be more efficient. Is the different with the experimental error? The standard deviation from replicates should be reported in the SI table including all the fitting results for quantitative comparison.

Now the supplementary table shows the very minor amplitude difference if comparing the absence of compounds to that of Ksg. The text also specifies that there are not substantial k_{off} differences (14 vs 11 s⁻¹). Thanks again for the meticulous verification.

Reviewer #4 (Remarks to the Author):

The authors have appropriately addressed my concerns and the manuscript is now ready for publication.